# Asymmetric random walks reveal that the chemotaxis network modulates flagellar rotational bias in *Helicobacter pylori*

**Jyot D Antani[1], Anita X Sumali[1], Tanmay P. Lele[2,3], Pushkar P. Lele[1]\***

[1]Artie McFerrin Department of Chemical Engineering, Texas A&M University, 3122 TAMU, College Station, Texas, 77843, USA; [2]Department of Biomedical Engineering, Texas A&M University, College Station, TX, 77840, USA; [3]Department of Translational Medical Sciences, Texas A&M University, Houston, Texas, 77030, USA

**Abstract** The canonical chemotaxis network modulates the bias for a particular direction of rotation in the bacterial flagellar motor to help the cell migrate toward favorable chemical environments. How the chemotaxis network in *Helicobacter pylori* modulates flagellar functions is unknown, which limits our understanding of chemotaxis in this species. Here, we determined that *H. pylori* swim faster (slower) whenever their flagella rotate counterclockwise (clockwise) by analyzing their hydrodynamic interactions with bounding surfaces. This asymmetry in swimming helped quantify the rotational bias. Upon exposure to a chemo-attractant, the bias decreased and the cells tended to swim exclusively in the faster mode. In the absence of a key chemotaxis protein, CheY, the bias was zero. The relationship between the reversal frequency and the rotational bias was unimodal. Thus, *H. pylori*'s chemotaxis network appears to modulate the probability of clockwise rotation in otherwise counterclockwise-rotating flagella, similar to the canonical network.

**\*For correspondence:**
plele@tamu.edu

**Competing interests:** The authors declare that no competing interests exist.

## Introduction

Over half of the human population is colonized by the motile gram-negative bacteria, *Helicobacter pylori*. *H. pylori* infections have been implicated in peptic ulcers as well as non-cardia gastric cancer (*Peek and Blaser, 2002*). Infections are promoted by the ability of the bacterium to swim with the aid of helical appendages called flagella (*Aihara et al., 2014*; *Ottemann and Lowenthal, 2002*). The flagellar filaments are rotated by transmembrane flagellar motors that repeatedly switch their direction of rotation. Owing to the unipolar location of the left-handed flagellar filaments (*Constantino et al., 2016*), counterclockwise (CCW) rotation of the motors causes the cell to run with the flagella lagging behind the body – a mode of motility termed as the *pusher* mode. The cell reverses with the body lagging the flagella when the motors switch the direction of rotation to clockwise (CW). This mode of motility is termed as the *puller* mode (*Lauga and Powers, 2009*). Modulation of the reversals between the two modes enables the cell to undergo chemotaxis — migration toward favorable chemical habitats (*Lertsethtakarn et al., 2011*; *Howitt et al., 2011*; *Johnson and Ottemann, 2018*). The core chemotaxis network is similar to that in *E. coli* (*Lertsethtakarn et al., 2011*; *Abedrabbo et al., 2017*; *Lertsethtakarn et al., 2015*; *Lowenthal et al., 2009b*; *Pittman et al., 2001*). Several components that form the flagellar motor are also similar to those in *E. coli* (*Lertsethtakarn et al., 2011*). How the chemotaxis network modulates flagellar functions in *H. pylori* remains unknown (*Lertsethtakarn et al., 2011*; *Lertsethtakarn et al., 2015*; *Jiménez-Pearson et al., 2005*).

Switching in the direction of flagellar rotation is promoted by the binding of a phosphorylated response regulator, CheY-P, to the flagellar switch. In the canonical chemotaxis network,

chemoreceptors sense extracellular ligands and modulate the activity of the chemotaxis kinase, which in turn modulates CheY-P levels. Increased CheY-P binding to the motor promotes CW rotation in an otherwise CCW rotating motor (*Sourjik and Wingreen, 2012*). The output of the flagellar switch is quantified by the fraction of the time that the motor rotates CW, termed CW$_{bias}$ (*Block et al., 1983*; *Yuan et al., 2012*; *Yang et al., 2020*). A decrease (increase) in CheY-P levels causes corresponding decrease (increase) in CW$_{bias}$ (*Cluzel et al., 2000*; *Scharf et al., 1998*). Thus, dynamic variations in the CW$_{bias}$ sensitively report changes in the kinase activity due to external stimuli as well as due to the internal noise in the network (*Cluzel et al., 2000*; *Korobkova et al., 2004*). In contrast, the frequency at which motors switch their direction does not accurately represent changes in the chemotaxis output as the frequency decreases when CheY-P levels increase as well as when CheY-P levels decrease (*Cluzel et al., 2000*). Prior work has focused on the effects of extracellular ligands on the frequency of cell reversals (*Collins et al., 2016*; *Machuca et al., 2017*; *Goers Sweeney et al., 2012*; *Rader et al., 2011*; *Sanders et al., 2013*; *Schweinitzer et al., 2008*) rather than the motor bias. It is unknown whether the chemotaxis network in *H. pylori* modulates the rotational bias.

In run-reversing bacteria, reversal frequencies can be readily quantified based on the number of reversals made by the swimming cell per unit time. To quantify the CW$_{bias}$, the duration for which the cell swims in each mode needs to be determined by distinguishing between the two modes. However, discriminating between the swimming modes of single *H. pylori* cells is difficult owing to the technical challenges in visualizing flagellar filaments in swimmers (*Constantino et al., 2016*; *Lowenthal et al., 2009a*). A popular method to quantify the CW$_{bias}$ is by monitoring the rotation of tethered cells, where a single flagellar filament is attached to a glass surface while the cell freely rotates (*Block et al., 1983*). Alternatively, the bias is determined by sticking the cell to the surface and monitoring the rotation of a probe bead attached to a single flagellar filament (*Ford et al., 2017*; *Yuan et al., 2010*). Such single motor assays have been employed successfully in *E. coli* because the filaments are spaced apart on the cell body. In *H. pylori*, however, the flagella are distributed in close proximity to one another at a single pole (*Qin et al., 2017*), increasing the likelihood of tethering more than one filament. Tethering of multiple flagella on the same cell eliminates rotational degrees of freedom, inhibiting motor function. Because of the limited measurements of the CW$_{bias}$, crucial features of the signaling network, such as the dynamic range of signal detection, adaptation mechanisms, and the roles of key chemotaxis-related proteins, remain unknown in *H. pylori*.

Here, we report a novel approach to measure the CW$_{bias}$ based on differences in the swimming speeds in the pusher and puller modes of *H. pylori*. We successfully employed this method to determine the effect of a known chemoattractant and varying temperatures on the CW$_{bias}$. Our observations suggest that the CW$_{bias}$ decreases upon stimulation with an attractant, similar to the canonical model. The default direction of flagellar rotation is CCW and the presence of CheY increases the CW$_{bias}$. The relationship between reversal frequencies and rotational bias is unimodal. These results are consistent with the notion that the chemotaxis network modulates the flagellar rotational bias (as well as the reversal frequencies) under environmental stimulus. Our quantitative model and simulations suggest that the basal chemotaxis output is likely tuned in *H. pylori* and other run-reversing bacteria to enhance diffuse spread. The approach discussed in this work provides a solid framework to study chemotaxis signaling and the behavior of the flagellar switch in *H. pylori*.

## Results

### Swimming speeds are asymmetric

To determine the behavior of flagellar motors in *H. pylori*, we tracked cell motility in the bulk fluid with a phase contrast microscope. The positions of single cells were quantitatively determined from digital videos with the aid of particle tracking (see Materials and methods). Owing to the use of low magnification microscopy, we were able to observe several cells exhibiting reversals in the field of view. A representative cell trajectory at 37°C is shown in *Figure 1A* (see another example in *Video 1*). With each reversal, the cell appeared to change from one mode of swimming to the other, although the modes could not be identified (as puller or pusher) because the flagella were not visible. Changes in the swimming modes were distinguished from rotational turns of the cell body – where

**Figure 1.** *H. pylori* swim forward and backward at different speeds. (**A**) Representative swimming trace of a single bacterium. Each reversal is represented by a filled circle. The beginning of the trajectory is denoted by an open circle. Uninterrupted swimming between two reversals was labeled as a segment and the segments were numbered chronologically. (**B**) The turn angles were exponentially distributed (n = 1653 samples); reversals mostly caused the cells to retrace their movements. (**C**) The swimming speed for a single cell over 1.5 s is indicated. The speeds alternated between high and low values with each reversal. Raw data is indicated in gray; filtered data is indicated in black. (**D**) The mean speed for each segment is indicated chronologically. Standard deviations are indicated. (**E**) The mean speed for the high (low) mode for each cell was calculated by averaging over all its high- (low-) speed segments. The distribution of the ratios of the high and low mean speeds for each cell is indicated. The mean ratio was 1.5 ± 0.4 (n = 250 cells).

The online version of this article includes the following source data for figure 1:

**Source data 1.**

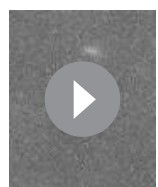

**Video 1.** A representative cell exhibits reversals within the field of view (movie has been slowed 3X).
https://elifesciences.org/articles/63936#video1

the swimming mode remains unchanged – by visually inspecting each reversal for each cell. The turn angle between the original direction just before and the new direction just after a reversal (ø) followed an exponential distribution with a peak ~180˚ (***Figure 1B***), indicating that cells simply retraced their paths for brief durations following each reversal. The flick of the flagellum that causes turn angles to be distributed ~90˚ in another run-reversing species *Vibrio alginolyticus* (***Stocker, 2011***), is unlikely to occur in *H. pylori*. The distance traveled between any two reversals was identified as a segment and numbered (***Figure 1A***). The swimming speeds over six consecutive segments are indicated in ***Figure 1C***.

The speeds were binned as per the segments, yielding $n+1$ bins for $n$ reversals. The mean speed from each bin was plotted for all the $n+1$ bins (*Figure 1D*). Mean speeds in alternate bins were anti-correlated: each reversal either decreased or increased the speed. This suggested that the speeds in the two modes were unequal. Such anti-correlation was consistently observed in a large population of cells (n = 250). The distribution of the ratio of their mean speeds in the fast and slow modes is shown in *Figure 1E*. The speed in the fast mode was ~1.5 times the speed in the slow mode.

## Cell swims faster in the pusher mode

A recent study attempted to visualize the flagella in *H. pylori* with high-magnification microscopy, and suggested that the cells swim faster in the pusher mode. However, the sample sizes were severely limited by the difficulties in visualizing flagella on the swimmers (*Constantino et al., 2016*). To conclusively determine the faster mode in *H. pylori*, we exploited the hydrodynamic coupling between swimmers and glass boundaries. Cells that swim very close to an underlying solid boundary exhibit circular trajectories owing to the increased viscous drag on the bottom of the cell and the flagellar filaments. CCW rotation of the left-handed helical filament causes the *pusher* to experience a lateral force that promotes CW circular tracks (*Figure 2A*, *DiLuzio et al., 2005*; *Lauga et al., 2006*). The situation is reversed when the filaments rotate CW. Thus, it is possible to discriminate between the two modes when a bacterium swims near a surface. We analyzed each cell that swam in circular trajectories near the surface and determined the mean speeds for the two directions. The cells were viewed from the bulk fluid, as indicated in *Figure 2A* (*right panel*). Four sample trajectories are shown in *Figure 2B*. For each cell, the CW trajectories were always faster relative to the CCW trajectories, indicating that the pusher mode was the faster mode (*Figure 2C*). This was confirmed over n = 116 cells; the mean ratio of the speeds of the CW trajectories to that of the respective CCW trajectories was ~1.6 ± 0.5 (*Figure 2D*).

## Partitioning of swimming speeds enables estimation of chemotaxis response to attractants

As *H. pylori* rotate their flagella CW in the puller mode, the $CW_{bias}$ could be calculated from the fraction of the time that the cells swam slower (see Materials and methods). This method worked for all the cells that reversed at least once in the field of view: the faster and slower modes could be discriminated from each other based on comparisons between the mean speeds before and after a reversal (as shown in *Figure 1D*). These cells consisted ~81% of the total data. The remaining cells did not reverse under observation; they persisted in a particular direction before exiting the field of view. Hence, these cells were termed as single-mode swimmers. As the mode of swimming could not be readily determined for these cells, those data were grouped into cells that swam near the surface for at least some time and those that did not. In the former group of cells, the majority was identified as pushers based on the direction of their circular trajectories near surfaces, as discussed in *Figure 2*. About 8% of the cells could not be identified and were excluded from the analysis. The distribution of the bias is shown in *Figure 3A*. The bias was similar to that observed in *E. coli* (*Block et al., 1983*; *Segall et al., 1986*; *Ford et al., 2018*; *Sagawa et al., 2014*; *Block et al., 1982*; *Stock et al., 1985*), suggesting that the basal chemotactic output in the two species is similar. As evident, most cells tended to rotate their motors CCW for a higher fraction of time.

Next, we imaged swimmers belonging to a *cheY*-deleted strain and observed that the trajectories of cells near a surface were exclusively CW circles. More than 150 cells were observed near surfaces and they exhibited CW trajectories. A fraction of the data is shown in *Figure 3B* and in *Appendix 1—figure 1*. This suggested that the default direction of flagellar rotation is CCW, similar to *E. coli* (*Ford et al., 2018*; *Liu et al., 2020*). Considering that the bias is zero in the absence and ~0.35 in the presence of CheY, CheY-P binding likely promotes CW rotation in an otherwise CCW rotating motor in *H. pylori*.

To test the idea that the chemotaxis network modulates the rotational bias, we employed our technique to quantify changes in the $CW_{bias}$ in swimmers when stimulated by a chemical attractant. We stimulated cells by adding them to a bath of urea (20 mM in motility buffer-MB, see Materials and methods), which is a potent chemoattractant for *H. pylori* (*Huang et al., 2015*). Following exposure to the attractant, the majority of the cells swam exclusively in the pusher mode – their post-stimulus $CW_{bias}$ was ~0 (*Figure 3C*). The reversal frequency also decreased in response to the

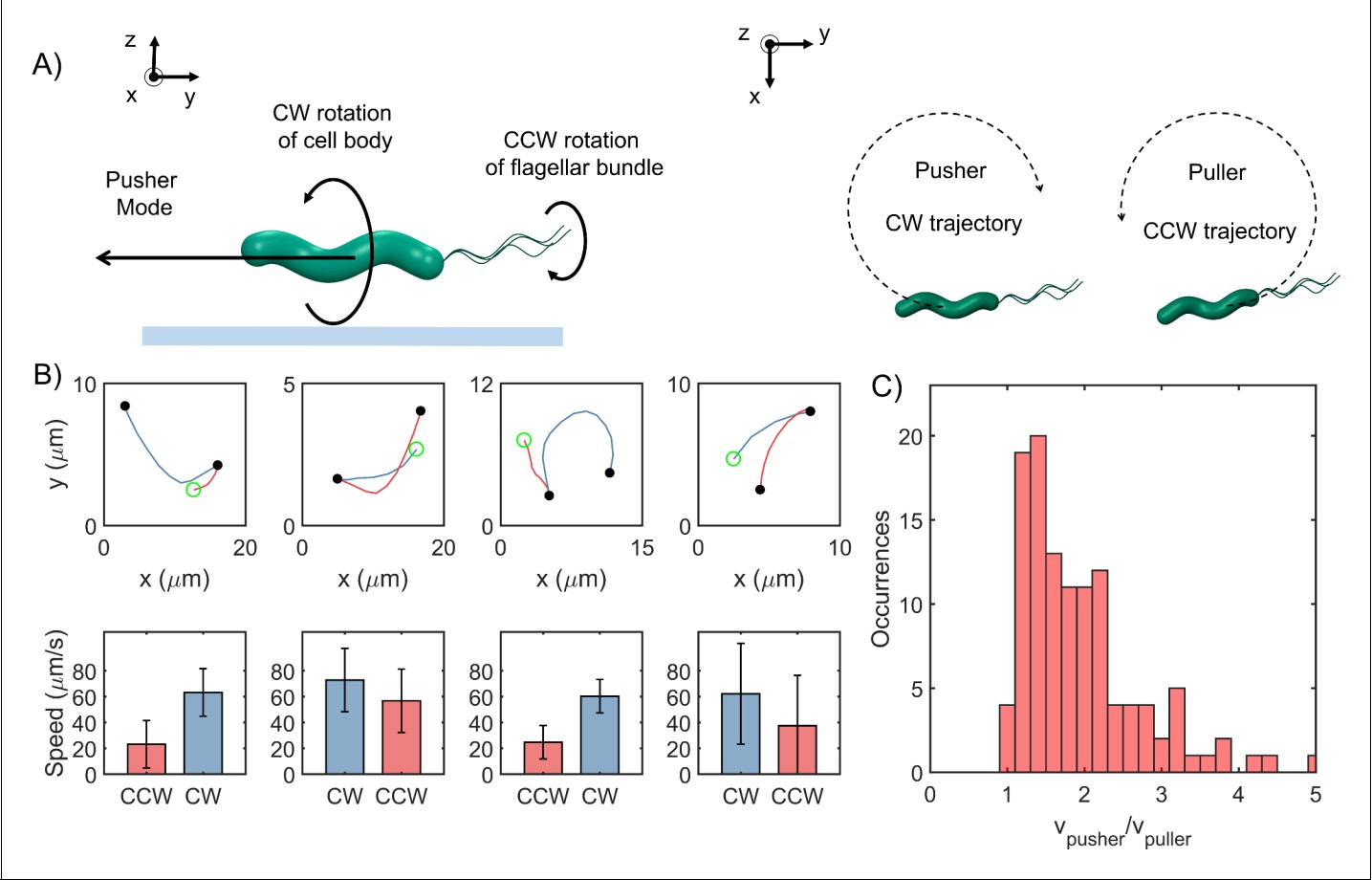

**Figure 2.** Cells swim faster in the pusher mode. (**A**) The viscous drag on the bottom of the cell body and the flagellar filament is higher near an underlying surface (indicated by the blue line in the *left panel*). The drag is lower on the top half of the body and filament. This difference in drag causes a lateral thrust on the cell, giving rise to circular trajectories: CW trajectory in the pusher mode and CCW trajectory in the puller mode (*right panel*). (**B**) *Top row:* Blue segments indicate CW trajectories; red segments indicate CCW trajectories. Filled circles indicate reversals; open circle indicates the beginning of the trajectory. *Bottom row:* The corresponding mean speeds and standard deviations are indicated for the two trajectories: CW tracks were always faster than CCW tracks. (**C**) The distribution of the ratio of the speeds along the pusher and puller modes is indicated (n = 116 cells). The mean ratio = 1.6 ± 0.5.

The online version of this article includes the following source data for figure 2:

**Source data 1.**

chemo-attractant (*Figure 3D*), which is in agreement with previous reports (*Machuca et al., 2017*; *Perkins et al., 2019*). In comparison, the post-stimulus $CW_{bias}$ in swimmers exposed to MB-only (control case) did not change significantly and continued to exhibit both modes of motility (*Figure 3C,D*). These observations are consistent with the notion that a reduction in the kinase activity upon the sensing of chemo-attractants inhibits the rotational bias of flagellar motors, similar to how the chemotaxis network modulates the response of *E. coli* to attractants (*Block et al., 1983*).

## Effect of thermal stimuli on chemotactic output

Several studies have characterized motility and chemotaxis in *H. pylori* at room temperatures (*Constantino et al., 2016*; *Howitt et al., 2011*; *Martínez et al., 2016*). Here, we explored how changes in the surrounding temperatures modulated the flagellar output in *H. pylori*. We recorded cell motility at different temperatures. The recording began ~5–10 min after each temperature change to provide adequate time for transient processes to stabilize (see Materials and methods for additional information). The mean pusher and puller speeds trended upwards with the temperature (*Figure 4A*, *left panel*), presumably through modulation of proton translocation kinetics that power

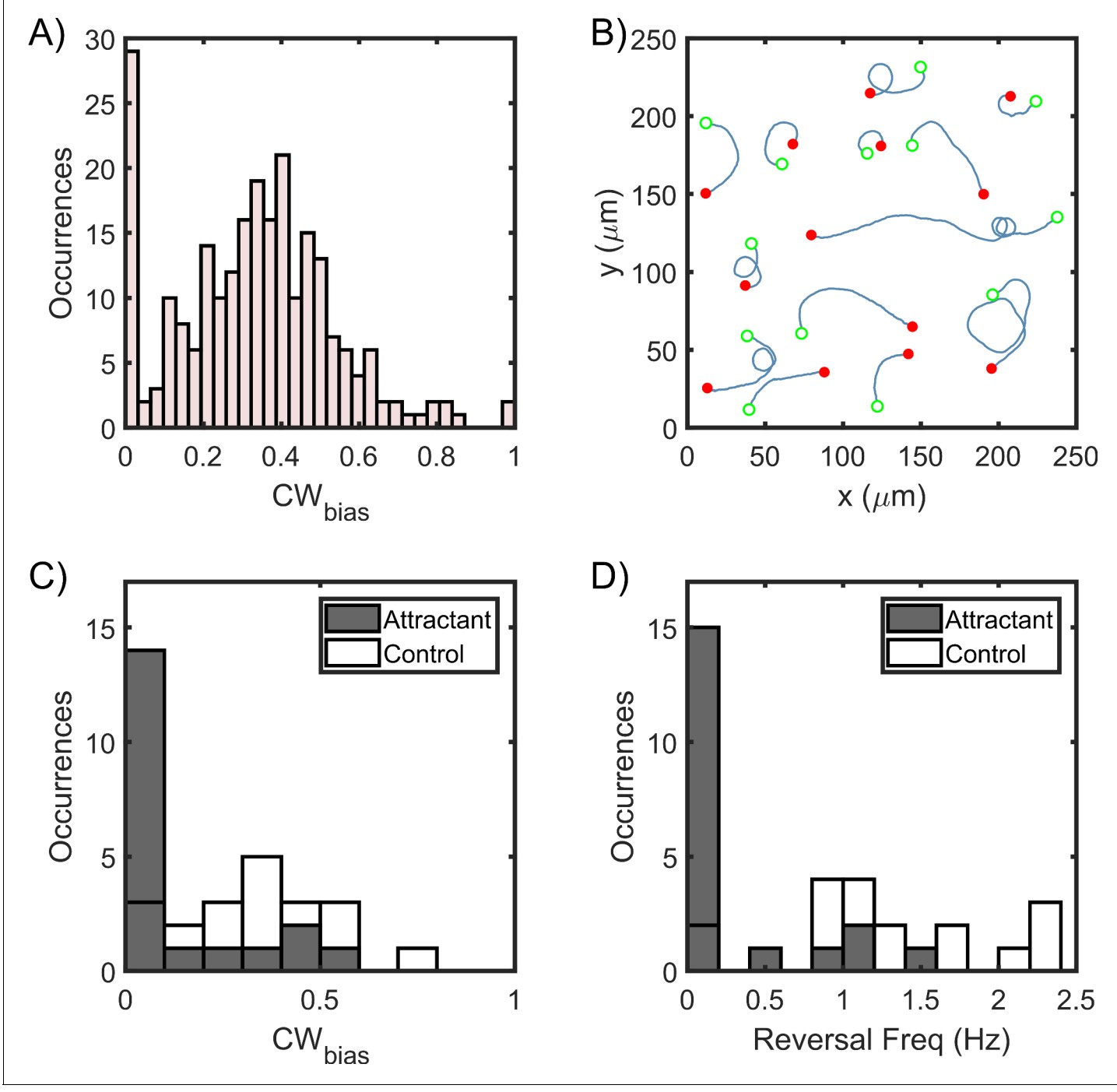

**Figure 3.** Asymmetric swimming speeds enable quantification of chemotaxis output. (A) $CW_{bias}$ was determined at 37°C in the absence of chemical stimulants. Cell trajectories with durations of 1 s or more were considered for calculation. The distribution was obtained from n = 240 cells. A Gaussian fit to the switching population (n = 212 cells) yielded $CW_{bias}$ = 0.35 ± 0.23 (mean ± standard deviation). (B) Single-cell trajectories of a Δ*cheY* mutant are indicated. Cells swam in CW-only trajectories, which indicate CCW flagellar rotation. Open green circles denote the start of a trajectory; filled red circles denote the end. The trajectories were spatially displaced to group them for the purpose of illustration and truncated to show the direction of rotation. Full trajectories and additional cells are included in ***Appendix 1—figure 1***. (C) The post-stimulus $CW_{bias}$ was monitored for ~30–60 s immediately following exposure to 20 mM urea (n = 20 cells); 14 cells swam exclusively in the pusher mode during the period of observation and displayed CW-only trajectories near surfaces. In the control case, cells were exposed to the buffer-only. The average post-stimulus $CW_{bias}$ was 0.31 ± 0.04 (mean ± standard error, n = 20 cells). The difference in the mean bias for the attractant and the control cases was significant (p-value<0.001). (D) The post-stimulus reversal frequency for cells treated with urea was 0.23 ± 0.09; those treated with the buffer had an average reversal frequency of 1.4 ± 0.04. The difference in the mean frequency for the attractant and the control cases was significant (p-value<0.001).

*Figure 3 continued on next page*

*Figure 3 continued*

The online version of this article includes the following source data for figure 3:

**Source data 1.**

the motor (*Yuan and Berg, 2010*). The ratio of the speeds in the two modes appeared to be independent of the temperature (*Figure 4A*, *right panel*). These responses are consistent with experiments in *E. coli* that show a strong influence of temperatures on the rotational speeds of the flagellar motor (*Yuan and Berg, 2010*; *Turner et al., 1996*; *Turner et al., 1999*).

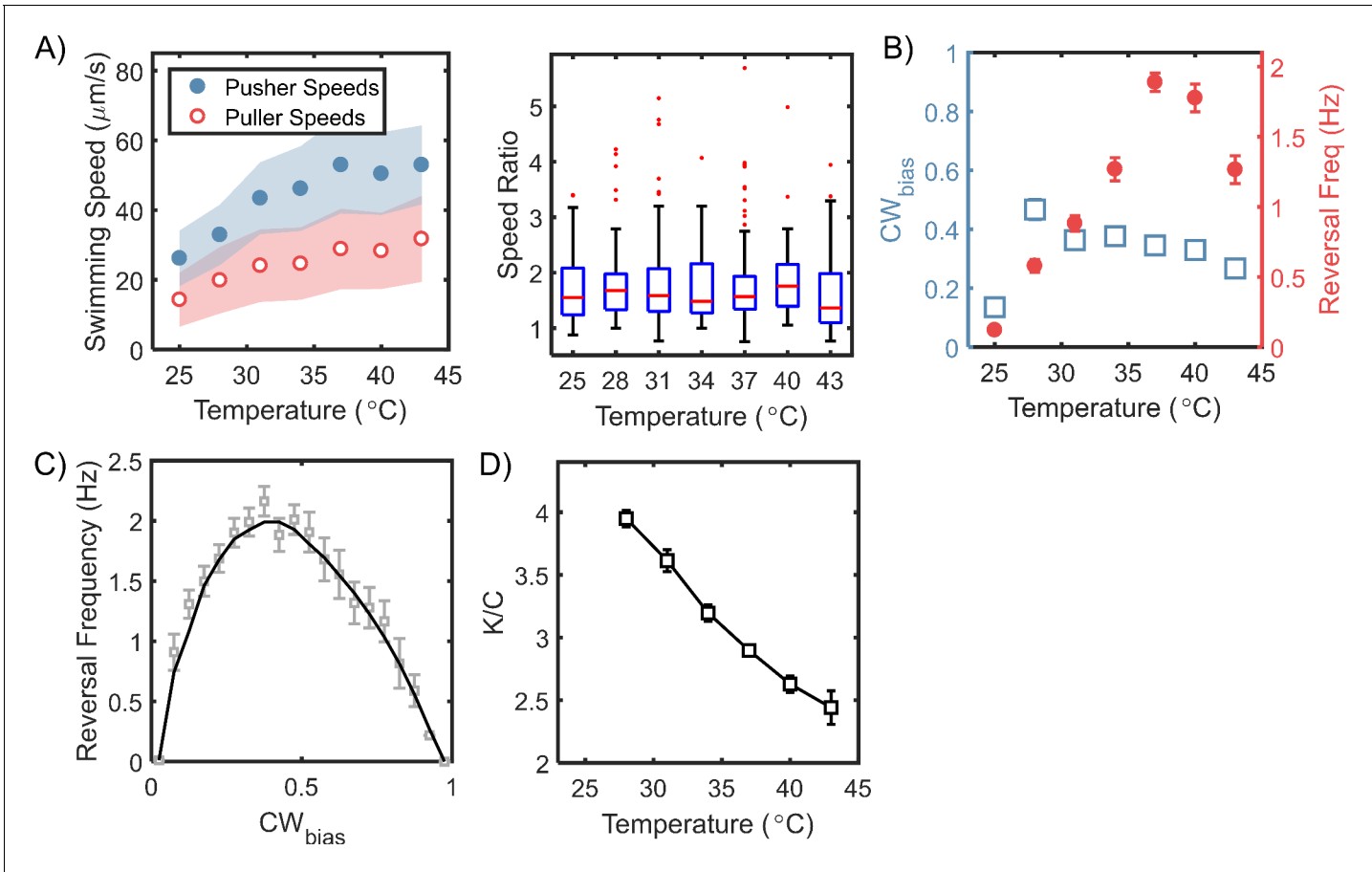

**Figure 4.** Steady-state chemotactic output is independent of temperature. (**A**) *Left*: Swimming speeds for each mode are plotted (mean ± standard deviation) for different temperatures. The speeds increased with temperature till 37˚C, after which they plateaued. The shaded regions indicate standard deviation. *Right*: The ratios of the pusher and puller speeds are independent of the temperatures, as indicated. A red horizontal line indicates the median ratio at each temperature, and the bottom and top borders of the encompassing box indicate the 25th and 75th percentiles. The extended lines span 99.3% of the data and the dots indicate outliers. (**B**) Mean $CW_{bias}$ (open squares) and mean reversal frequencies (filled circles) are plotted over a range of temperatures. The switching frequency was at a maximum at the physiological temperature (37˚C) and decreased at higher and lower temperatures. The $CW_{bias}$ increased with the temperature and plateaued above 30˚C. The mean values are indicated with standard error. Each data-point was averaged over n ≥ 80 cells. (**C**) The relationship between reversal frequency and $CW_{bias}$ is indicated. The values were obtained from the combined datasets over the entire range of temperatures that we studied (n = 972 cells). The $CW_{bias}$ was binned (bin size = 0.05), and the mean reversal frequency for each bin was estimated. The mean and standard errors are indicated in grey. The black curve is a guide to the eye. (**D**) The estimated ratio of the CheY-P dissociation constant (*K*) and the intracellular CheY-P concentrations (*C*) is indicated as a function of the temperature. The ratios were calculated from the data in (**B**) following a previous approach (*Turner et al., 1999*). The number of binding sites for CheY-P in *H. pylori* ~ 43 was estimated from the relative sizes of the flagellar C-ring (see Appendix 2 and *Qin et al., 2017*). The ratio of the dissociation constants for the CCW and the CW motor conformations was assumed to be similar to that in *E. coli* (~ 4.7 from *Fukuoka et al., 2014*).

The online version of this article includes the following source data for figure 4:

**Source data 1.**

The frequency of reversals increased steadily with temperature up to 37°C, whereas the steady-state $CW_{bias}$ varied weakly with temperature (**Figure 4B**). At room temperatures, the $CW_{bias}$ was the lowest, indicating that the cells mostly prefer to swim in the pusher mode. Next, we combined our data over the entire range of temperatures (25–43°C) and for each cell, plotted the reversal frequency against its $CW_{bias}$. The reversal frequency was maximum at ~0.4  $CW_{bias}$ and decreased on either side (**Figure 4C**); also see **Appendix 3—figure 1**. As our results suggest that CheY-P binding increases the $CW_{bias}$ (**Figure 3B,C**), this also means that the reversal frequency has a similar unimodal dependence on CheY-P levels. Hence, we propose that changes in the reversal frequency in *H. pylori* cannot provide accurate information about the effect of stimulants on the kinase activity (i.e. whether a stimulant increases or decreases the activity). On the other hand, the rotational bias is likely a better measure of the kinase activity, similar to that in *E. coli*.

In *E. coli*, flagellar switching has been well described by a two-state model, where the binding of phosphorylated CheY (CheY-P) to the flagellar switch stabilizes the CW conformation (**Turner et al., 1999**). In the absence of CheY-P, the probability of observing CW rotation in an otherwise CCW-rotating motor decreases with increasing temperatures (**Turner et al., 1996**). The chemotaxis network itself adapts such that the steady-state levels of CheY-P are independent of the temperatures (**Paulick et al., 2017**). Assuming that CheY-P levels are also independent of the temperature in *H. pylori*, the relative insensitivity of the rotational bias in **Figure 4B** suggested that the dissociation constant for CheY-P/switch interactions likely decreased with rising temperatures. Following the thermodynamic analysis of Turner and co-workers for a two-state flagellar switch (**Turner et al., 1999**), we calculated the dissociation constant normalized by CheY-P levels, as shown in **Figure 4D** (see Appendix 2 for details). Assuming that the CheY-P levels are ~3 µM (**Cluzel et al., 2000**), we estimate the dissociation constant to be ~9 µM at 37°C.

## Speed asymmetry promotes diffusion

Even without chemotaxis, motility enhances the spread of bacteria, lending a significant advantage over immotile bacteria in exploring three-dimensional spaces (**Josenhans and Suerbaum, 2002**). Bacterial motion becomes uncorrelated over long times and large length-scales in the absence of a signal. Several previous works have modeled the diffusion of motile bacteria by assuming that the reversal wait-times are exponentially distributed (**Berg, 1993**; **Lovely and Dahlquist, 1975**; **Lauga, 2016**; **Theves et al., 2013**). The wait-time refers to the time between two consecutive reversals. In some bacterial species that exhibit runs and reversals, the wait-time is Gamma distributed (**Theves et al., 2013**; **Morse et al., 2016**; **Xie et al., 2011**). The assumption of exponentially distributed wait-times leads to inaccurate predictions in such species (**Theves et al., 2013**).

Our cell-tracking analysis revealed that the reversal wait-times were Gamma distributed in *H. pylori* (**Figure 5A**, also see separate distributions for the two swimming modes in **Appendix 3—figure 2**). When calculating the wait-times, we excluded the beginning of each cell-trajectory just before the first reversal and the end of each cell-trajectory just after the final reversal. To derive an explicit expression for the diffusivity of asymmetric run-reversers that exhibit Gamma distributed reversal intervals, we preferred to modify a previous approach developed for symmetric run-reversers (**Großmann et al., 2016**) rather than a more general model (**Detcheverry, 2017**) – see Appendix 4 for details. Briefly, the velocities of a bacterium that swims at $v_0$ µm/s in its slower mode was expressed as: $\mathbf{v}(t) = v_0\, h(t)\, [1 + aH(h)]\mathbf{e}(t)$. The direction of swimming was described by the function $h(t)$, which alternated between +1 and −1 with each reversal (**Figure 5B**). A Heaviside function, $H(h)$ and the asymmetry parameter, $a$, characterized the magnitudes of the speeds in the two directions: $v_0$ and $a$. The $CW_{bias}$ was assumed to be constant (= 0.5) for simplicity.

The deviation of the cell from a straight line during a run (or reversal) occurred due to rotational diffusion, described by $\frac{d\theta}{dt} = \sqrt{2D_\theta}\xi(t)$. White noise characteristics were $\langle\xi(t)\rangle = 0$, and $\langle\xi(t)\xi(t + \tau)\rangle = \delta(\tau)$, where $D_\theta$ is the rotational diffusion coefficient. Another randomizer of the bacterial walk is the turn angle, $\emptyset$, which is the angle between the original direction just before and the new direction just after a reversal. The turn angle is likely influenced by kinematic properties: cell shape, filament bundling dynamics, and the flexibility of the flagellar hook. After taking into account the specific form of the reversal wait-time distribution for *H. pylori* (**Figure 5A**), we obtained the following expression for the asymptotic diffusion coefficient from the velocity correlation over long-times:

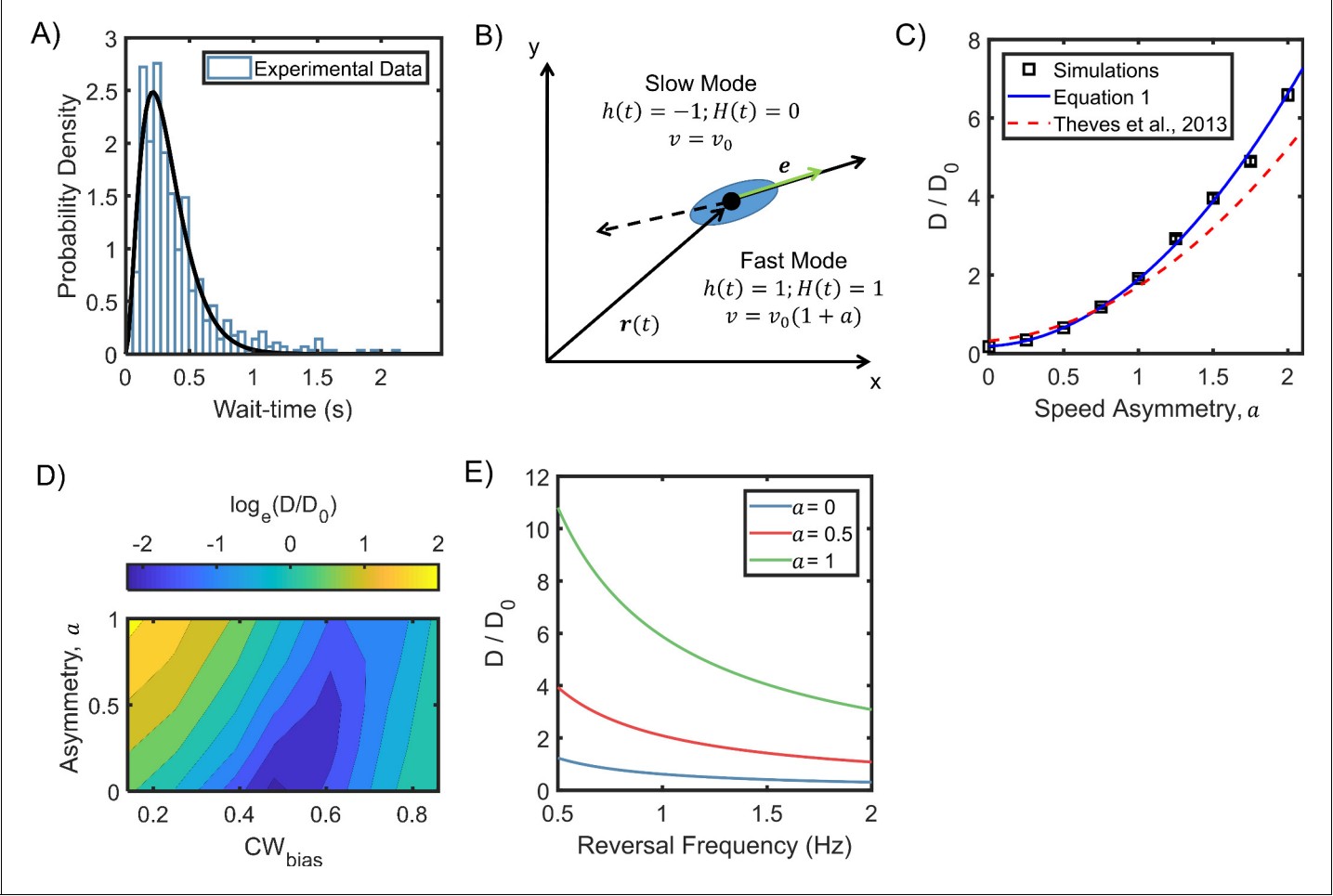

**Figure 5.** Asymmetric random walks in a run-reversing bacterium. (A) Experimentally observed wait-time intervals for runs and reversals obey a Gamma distribution (n = 515 samples): the shape and scale parameters were $k = 2.92 \pm 0.06$ and $\theta = 0.11 \pm 0.00$, respectively. (B) Cell swims at $v_0$ μm/s in the puller (slower) mode, and at $v_0(1 + a)$ μm/s in the pusher (faster) mode. The symmetric case is described by $a = 0$, where the run and reverse speeds are equal. Cell alignment is described by the unit vector $e$. (C) The diffusion coefficients predicted from equation 1 are indicated as a function of the asymmetry in speeds (blue curve). An alternate model that assumes exponentially distributed wait-time intervals in asymmetric swimmers under predicted the diffusivity, as shown by the dotted curve (*Theves et al., 2013*). Symbols indicate coefficients calculated from simulation runs (see Appendix 5). The parameters were based on experimental measurements: mean wait-time = 0.3 s, $\alpha = 0.86$, and $v_0 = 25$ μm/s. $D_\theta = 0.02$ s$^{-1}$ from (*Großmann et al., 2016*). Diffusion coefficients have been non-dimensionalized with $D_0 = v_0^2/3\omega_p$ (*Lovely and Dahlquist, 1975*), where $\omega_p$ is the mean reversal frequency at the physiological temperature (*Figure 4B*). (D) Diffusion coefficients were calculated from simulations of cell motility in the absence of a stimulus over a range of $a$ and CW$_{bias}$ values (see Appendix 5 for details). The diffusion coefficients were normalized with $D_0$. The sum of the mean wait-times (CW and CCW) was fixed at 0.35 s. (E) Predicted diffusivity is indicated over a range of typical reversal frequencies. Here, $\alpha = 0.86$ and $D_\theta = 0.02$ s$^{-1}$.

The online version of this article includes the following source data for figure 5:

**Source data 1.**

$$D = \frac{v_0^2}{2D_\theta}\left[\frac{(1+a)^2+1}{2}\cdot\left\{\begin{array}{c}1-\frac{\omega}{D_\theta}\left(1-\frac{(3\omega)^3}{(3\omega+D_\theta)^3}\right)\\+\frac{|\langle cos\varnothing\rangle|^2\omega}{D_\theta}\cdot\frac{(3\omega)^3}{(3\omega+D_\theta)^3}\cdot\frac{\left((3\omega+D_\theta)^3-(3\omega)^3\right)^2}{(3\omega+D_\theta)^6-|\langle cos\varnothing\rangle|^2(3\omega)^6}\end{array}\right\}-(1+a)\left\{\frac{|\langle cos\varnothing\rangle|\cdot\omega}{D_\theta}\cdot\frac{\left((3\omega+D_\theta)^3-(3\omega)^3\right)^2}{(3\omega+D_\theta)^6-|\langle cos\varnothing\rangle|^2(3\omega)^6}\right\}\right]$$

(1)

The reversal frequency is indicated by ω. The expression correctly reduces to that for the symmetric swimmer (*Großmann et al., 2016*), for α $(= |\langle cos \emptyset \rangle|)$ = 1, and $a$ = 0.

As shown in *Figure 5C*, the diffusion coefficients increased with the asymmetry-parameter, $a$. As per the predictions, asymmetric run-reversers ($a \neq 0$) spread faster than symmetric run-reversers ($a$ = 0). Next, we carried out stochastic simulations of 1000 cells that underwent asymmetric run-reversals with Gamma distributed intervals (see Appendix 5). The diffusion coefficients from the simulations matched predictions from our model that incorporated Gamma distributed wait-times. Having validated our simulations, we estimated the diffusion coefficients for arbitrary $CW_{bias}$ values over varying $a$. As shown in *Figure 5D*, the simulated diffuse spread was low when cells covered similar distances in the forward and backward directions, thereby minimizing net displacement. This tended to occur for swimmers with low $a$ values that swam for equal durations in the two directions ($CW_{bias}$ ~ 0.5). For any given $a$, the diffuse spread increased with the net displacement during a run-reversal, for example, when the swimmer preferred the slower mode much more than the faster mode. The net displacement, and hence, the spread tended to be the highest when the cells spent a greater fraction of the time swimming in the faster mode compared to the slower mode. Thus, in *H. pylori*, the tendency to spend more time in the faster pusher mode (basal $CW_{bias}$ ~ 0.35, *Figure 3A*) is advantageous (*Figure 5D*). This advantage is amplified by increasing pusher speeds relative to the puller speeds. However, a very low basal value of the $CW_{bias}$ is disadvantageous from a chemotaxis perspective. *H. pylori* appear to respond to attractants by reducing their $CW_{bias}$ (*Figure 3C*). They would lose their ability to respond to attractants if the pre-stimulus (basal) bias was close to its minimum value (=0). It is possible, therefore, that the basal activity of the chemotaxis network is optimized in asymmetrically run-reversing bacteria to promote higher diffusive spread while retaining the ability to respond to chemical stimuli.

Finally, longer durations of runs and reversals helped cells cover larger distances. Thus, the diffusion coefficient was inversely dependent on the run-reversal frequency (*Figure 5E*). As the reversal frequencies reach a maximum at 37°C (*Figure 4B*), it is possible that cells at physiological temperatures spread slower in a niche over long times, providing more time for cells to adhere to surfaces.

## Discussion

*H. pylori* experience physiological temperatures (~37°C) in their human hosts. Here, we characterized flagellar functions at physiologically relevant temperatures. Our experiments with a mutant lacking *cheY* showed that the flagellar motors in *H. pylori* rotate CCW by default (*Figure 3B*). At native CheY-P levels, motors in wild-type cells spent about 35% of the time rotating CW (*Figure 3A*) – thus, the probability of CW rotation increases with CheY. Our experiments further showed that treatment of wild-type cells with a potent attractant (urea) decreased the rotational bias ($CW_{bias}$). These results are consistent with a model in which the chemotaxis network controls the levels of CheY-P to modulate the probability of CW rotation in an otherwise CCW rotating flagellar motor. If so, then the chemotaxis networks in the two species, *E. coli* and *H. pylori*, modulate flagellar functions in a similar manner.

Earlier works focused exclusively on the effect of chemoeffectors on steady-state reversal frequencies in *H. pylori* to characterize chemotaxis responses (*Collins et al., 2016*; *Machuca et al., 2017*; *Goers Sweeney et al., 2012*; *Rader et al., 2011*; *Sanders et al., 2013*; *Schweinitzer et al., 2008*). Because diffusion scales inversely with the reversal frequency (*Figure 5E*), increases in frequency might help a cell linger in a niche. However, mere variations of the steady-state reversal frequencies with the local stimulant concentrations (or temperatures) does not enable chemotaxis (*Berg, 1993*). By combining data collected over a range of temperatures, we showed that the dependence of the reversal frequency on the rotational bias is unimodal (*Figure 4C*). This means that the reversal frequency does not have a unique value with respect to the rotational bias (other than at maximal frequency), similar to that in *E. coli* (*Figure 6A*). Hence, the reversal frequency value is also unlikely to be unique with respect to the kinase activity (the corresponding relationship in *E. coli* is depicted in *Figure 6B*). Therefore, changes in the reversal frequencies by themselves are unlikely to accurately report changes in the chemotaxis output in *H. pylori*. Our results suggest that the rotational bias must be quantified to accurately determine the chemotaxis output in *H. pylori*.

Our model for diffusive spread of motile bacteria indicated that run-reversing bacteria that undergo asymmetric random walks diffuse faster than symmetric run-reversers (*Figure 5C*). This is

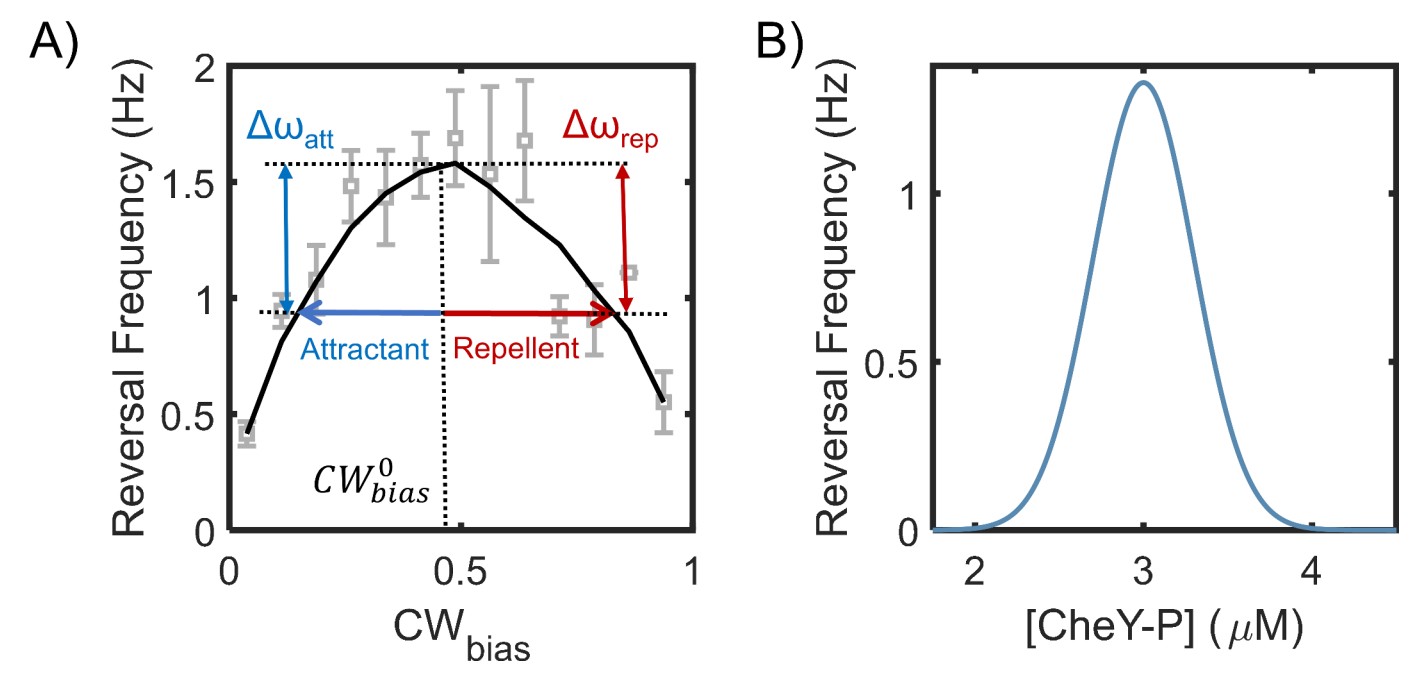

**Figure 6.** Motor reversal (switching) frequencies versus $CW_{bias}$ and CheY-P. (**A**) The dependence of motor reversal frequencies in *E. coli* on the $CW_{bias}$ is unimodal (***Montrone et al., 1998***), similar to *H. pylori* (***Figure 4B***). The symbols indicate experimental data from ***Montrone et al., 1998***. The black curve is a guide to eye. The blue and red arrows indicate the effect of attractants and repellents on the $CW_{bias}$, respectively. The corresponding changes in the reversal frequency are similar ($\Delta\omega_{att} \sim \Delta\omega_{rep}$). (**B**) The dependence of switching frequency on CheY-P levels is also unimodal in *E. coli* (***Cluzel et al., 2000***). Thus, an attractant as well as a repellent can induce a drop in the frequency.

expected, as symmetric run-reversals tend to minimize net displacements. Simulations of bacterial diffusion in the absence of stimulants indicated that the diffusive spread is higher in asymmetric run-reversers when the cells spend a greater fraction of the time swimming in the faster mode compared to the slower mode. Thus, the preference for the faster pusher mode (lower $CW_{bias}$) in *H. pylori* is advantageous as it helps them spread faster (***Figure 5D***). However, *H. pylori* appear to respond to attractants by reducing their $CW_{bias}$ (***Figure 3C***). A very low value of the basal bias would inhibit the ability to respond to attractants entirely. Hence, we propose that the basal activity of the chemotaxis network is probably tuned to promote higher diffusive spread while optimizing chemotaxis performance. In general, asymmetry in swimming – differences in swimming speeds or differences in the amount of time spent in any one mode or both – may provide evolutionary benefits to run-reversing bacteria by enhancing their spread.

The response of the chemotaxis network to external stimuli is conventionally measured by determining the rotational bias (***Block et al., 1983***; ***Yang et al., 2020***; ***Lele et al., 2015***; ***Jasuja et al., 1999***). Tethering cells to glass surfaces is the preferred method of determining the rotational bias. This approach is only useful when one can ascertain that the filament has adhered to the surface, for example with the use of antiflagellin antibodies that irreversibly link the filament to the surface. However, some studies may forego the use of antibodies when determining the bias. This is problematic as the cell can appear to be tethered but instead it pivots about its non-flagellated pole on a surface while the free rotation of the invisible filament causes the cell to rotate. This can lead to the mischaracterization of the direction of flagellar rotation, and therefore the rotational bias (***Dominick and Wu, 2018***; ***Lele et al., 2016***; ***Chawla et al., 2020***). Alternately, the signaling output has been determined via Förster resonance energy transfer-based measurements of in vivo enzymatic reactions (***Sourjik et al., 2007***). But, neither of these approaches has been realized in *H. pylori*. Here, we characterized the rotational bias based on the asymmetry in the swimming speeds. Our use of low-magnification microscopy allowed us to collect large sample sizes to characterize flagellar functions, considerably improving on earlier efforts (***Constantino et al., 2016***).

To prevent the cell from tumbling during a reversal, all the flagellar motors in a single cell of *H. pylori* must switch synchronously from one direction to the other. Indeed, tumbles were rarely observed. The most frequent turn angles were ~180°, which confirmed that the cells retraced their paths following a reversal – this would not have been the case if only a fraction of the motors switched to the opposite direction. This makes our approach feasible for determining the $CW_{bias}$ for an individual cell from its swimming speeds – which reflects the collective action of all the motors – rather than sampling individual motors. How are such multiple stochastic switchers coupled in *H. pylori*? One possibility is that the flagellar switch in *H. pylori* is ultrasensitive to small fluctuations in CheY-P levels, similar to the switch in *E. coli* (*Cluzel et al., 2000*). The close proximity of the multiple motors at a single pole in *H. pylori* also means that the local concentration of CheY-P in the vicinity of each flagellar switch is similar. This increases the probability of concerted switching in all the motors.

In *V. alginolyticus,* asymmetry in swimming speeds has been observed only near bounding surfaces but not in the bulk fluid (*Magariyama et al., 2005*). A limitation of our method is that it is unsuitable for tracking chemotaxis response dynamics in such species, as the asymmetry is lost whenever the cells migrate away from surfaces. In *H. pylori*, although, we observed asymmetric speeds in some cells even at a separation of ~200 µm from any bounding surfaces (see Appendix 6), similar to *Pseudomonas putida* (*Theves et al., 2013*). Therefore, the asymmetry is unlikely to be a surface-effect in *H. pylori*. The effect could be due to differences in the flagellar shape and forms (*Kinosita et al., 2018*) or the swimming gait in the pusher and puller modes (*Lele et al., 2016*; *Liu et al., 2014*). It is more likely that the asymmetry in speeds arises due to the differences in the CW and CCW flagellar rotational speeds, as is the case with *E. coli* – which run and tumble – and *Caulobacter cresecentus* (*Yuan et al., 2010*; *Lele et al., 2016*) – which exhibit symmetric swimming speeds in the pusher and puller modes (*Table 1*). Such differences in the speeds at which motors rotate CW and CCW depend on the external viscous loads (*Yuan et al., 2010*; *Lele et al., 2016*). It is possible therefore, that the asymmetry in *H. pylori* is also load-dependent; vanishing for longer filament lengths in highly viscous microenvironments or for very short filaments. The asymmetry is further expected to depend sensitively on the expression of the flagellar genes, which is modulated by environmental conditions (*Spohn and Scarlato, 1999*). The asymmetry was prominently observable in our work with a careful control of experimental conditions (Materials and methods).

The flagellar motors in *H. pylori* and *E. coli* share structural similarities and have several orthologous components. The core chemotaxis network in the two species is also similar with the exception of a few enzymes (*Lertsethtakarn et al., 2011*; *Howitt et al., 2011*; *Lertsethtakarn et al., 2015*; *Jiménez-Pearson et al., 2005*). CheY, in its phosphorylated form, modulates flagellar functions in both species by interacting with components of the flagellar switch (*Lertsethtakarn et al., 2011*; *Lowenthal et al., 2009a*; *Qin et al., 2017*; *Lam et al., 2010*). Our results suggest that the regulatory function of CheY-P is also similar in the two species, that is, CheY-P binding to the motor increases the probability of CW rotation. If so, then the implications of this finding are significant. Because *H. pylori* can retrace their paths upon a reversal unlike *E. coli*, modulation of the rotational bias is bound to undermine chemotaxis when the cell enters the puller mode. Then, the cell would likely need a mechanism to rectify its movements with respect to the source or some type of feedback between the motors and the receptors to successfully migrate in response to chemical gradients. We anticipate that the approaches described in this work will help uncover these mechanisms and

**Table 1.** Speed asymmetry across different bacterial species.

| Species | Swimming Speed Ratio | Motor Speed Ratio | Reference |
|---|---|---|---|
| *H. pylori* | 1.5 | - | This work |
| *P. putida* | 2 | - | *Theves et al., 2013* |
| *V. alginolyticus* | 1.5 | - | *Magariyama et al., 2005* |
| *Burkholderia* spp. | 3.9 | - | *Kinosita et al., 2018* |
| *Vibrio fischeri* | 3.4 | - | *Kinosita et al., 2018* |
| *C. crescentus* | 1 | ~2 | *Lele et al., 2016*; *Liu et al., 2014* |
| *E. coli* | ~1.3 | 1.3 | *Yuan et al., 2010*; *Lele and Berg, 2015* |

identify unknown protein functions. Our approach is extensible to any run-reversing species that exhibit asymmetric swimming speeds, paving the way to study signaling dynamics in other run-reversing bacterial species.

# Materials and methods

**Key resources table**

| Reagent type (species) or resource | Designation | Source or reference | Identifiers | Additional information |
|---|---|---|---|---|
| Cell line (*H. pylori*) | PMSS1 | Ottemann Lab *Arnold et al., 2011* | | |
| Chemical compound, drug | Brucella Broth | Millipore Sigma | B3051 | |
| Chemical compound, drug | Columbia agar | Thermo Scientific Oxoid | CM0331 | |
| Chemical compound, drug | Defibrinated Horse Blood | Hemostat Laboratories | DHB100 | |
| Chemical compound, drug | Fetal Bovine Serum | Gibco | 10438 | |
| Chemical compound, drug | Polymixin-B sulfate | Alfa Aesar | J6307403 | |
| Chemical compound, drug | Vancomycin hydrochloride | Sigma Aldrich | V1130 | |
| Chemical compound, drug | β-Cyclodextrin | Sigma Aldrich | C4767 | |
| Chemical compound, drug | Urea | Fisher Scientific | BP169 | |

## Strains and cell culturing

All the work were done with *H. pylori* PMSS1. Cultures of microaerophilic *H. pylori* were grown in an incubator with controlled temperature and $CO_2$ environment (Benchmark Incu-Shaker Mini $CO_2$). The incubator was maintained at 10% $CO_2$, 37°C. Fresh colonies were streaked out before each experiment on Columbia agar plates supplemented with 2.5 units/mL Polymixin-B, 10 μg/mL Vancomycin, 2 mg/mL β-cyclodextrin, and 5% v/v defibrinated horse blood. Colonies appeared on the horse-blood agar plates within 3–4 days and were picked with the aid of sterilized cotton-tipped applicators. The cells were then inoculated in 5 mL of BB10 (90% Brucella Broth + 10% Fetal Bovine Serum) to grow overnight cultures. No antibiotics were added to the liquid cultures as per previous protocols (*Machuca et al., 2017*; *Huang et al., 2017*). Overnight cultures were grown for ~16 hr to an $OD_{600}$~0.25–0.5 and diluted to $OD_{600}$~0.1 in fresh BB10. The day cultures were grown to an $OD_{600}$~0.125–0.15 in the shaker incubator set at 170 rpm under 10% $CO_2$ and at 37°C. Prior to imaging, the cells were diluted in a motility buffer (MB- 0.01 M phosphate buffer, 0.067 M NaCl, and 0.1 mM EDTA, pH~7.0) at ~6–7% v/v (BB10/MB).

## Motility assays

Cells were imaged in a culture-dish (Delta T system, Bioptechs Inc) on a phase-contrast microscope (Nikon Optiphot) equipped with a 10X phase objective. The dish was kept covered with a lid that was not airtight and that allowed a part of the top liquid surface to be exposed to air. Videos were recorded with a CCD camera (IDS model UI-3240LE) at 45 frames per second. Unless otherwise specified, the objective was focused ~5–20 μm away from the bottom surface of the culture-dish. All experiments were performed at 37°C unless otherwise noted.

## Temperature control

The microscope was housed inside a temperature control chamber (ETS Model 5472, Electro-Tech Systems, Inc), which enabled precise control over the temperature during the experiments. The grown cultures were stored in flasks within the chamber. Prior to each measurement, ~50 μL of cells were diluted in ~1.3–1.5 mL of MB. The entire mixture was then transferred to the culture dish and covered with the lid. As the cell density was low (~$4 \times 10^6$ cells/mL) and as the liquid surface was exposed to air, oxygen gradients were minimized; the cells remained motile in MB for over an hour.

In the case of the temperature variation experiments, the cells were visualized in the dish ~5–10 min after each change in the temperature. Once recording was completed, the contents of the culture dish were emptied. The dish was then flushed with ethanol followed by copious amounts of DI

water outside the chamber. The dish was then reused for further experiments. The whole cycle was repeated each time the temperature was changed.

## Chemoattractant response

We filled the culture dish with 20 mM urea (Fisher Chemical) in MB at 37°C, which served as an attractant. In the control case, no urea was added to the MB in the dish. We pipetted 50 μL of the cell culture into the dish prior to imaging. Videos were recorded and analysis was performed on the videos once the hydrodynamic flows visually subsided (~30 s).

## Data analysis

The low cell density enabled us to employ particle-tracking methods to record the swimming trajectory of each cell (**Ford et al., 2017**). All the videos were analyzed with custom-written MATLAB codes based on centroid-detection-based particle-tracking routines (**Crocker and Grier, 1996**). The experimental data shown in **Figure 3C,D** and **Figure 4** were obtained from two biological and multiple technical replicates. All other data were collected from five or more biological and multiple technical replicates.

## CW_bias calculations

Recorded videos were visually scanned with ImageJ (NIH) to confirm the number of reversals for each cell. The distance traveled between any two reversals was identified as a segment and numbered (see **Figure 1A**). The speeds were binned as per the segments, yielding *n+1* bins for *n* reversals. A reversal changes the mode of motility between the pusher and the puller mode. On the other hand, a 180° turn by the cell maintains the same mode. Each reversal was therefore confirmed visually to distinguish between reversals; 180° turns were rarely observed. In cells that swam near surfaces, the pusher and puller modes were readily determined as described in **Figure 2**. In cells that did not swim near surfaces, we compared the mean speeds, which alternated as shown in **Figure 1D**. All the *alternating* fast speed-bins were labeled as pushers; *alternating* low speed-bins were labeled as pullers. The video frames corresponding to the puller bins were labeled as puller frames.

To determine the CW_bias, cells that were observed for at least 0.5 s were retained for analysis. CW_bias was calculated as the fraction of the time that a cell swam in the puller (slower) mode, which corresponds to CW rotation of the filament. To do this, the number of frames in which the $i^{th}$ cell swam in the puller mode (i.e. puller frames), $N_i^{CW}$, was divided by the total frames over which the cell was observed, $N_i$, to yield:

$$CW_{bias,i} = \frac{N_i^{CW}}{N_i}$$

The error associated with the calculation of $CW_{bias,i}$ values decreases with increasing $N_i$. But, different cells were observed for different durations; hence the $CW_{bias,i}$ values were allocated weights that corresponded to their respective durations: $W_i = \frac{N_i}{\sum N_i}$. Mean bias was determined as:

$$CW_{bias} = \sum W_i CW_{bias,i}$$

Reversal frequency was determined in a similar manner.

## Acknowledgements

We thank Professor Karen Ottemann for the PMSS1 strains, and Professor Michael Manson and Rachit Gupta for comments.

## Additional information

### Funding

| Funder | Grant reference number | Author |
|---|---|---|
| Cancer Prevention and Research Institute of Texas | RP170805 | Pushkar P Lele |
| National Institute of General Medical Sciences | R01-GM123085 | Pushkar P Lele |
| Cancer Prevention and Research Institute of Texas | RR200043 | Tanmay, P. Lele |

The funders had no role in study design, data collection and interpretation, or the decision to submit the work for publication.

### Author contributions

Jyot D Antani, Data curation, Software, Formal analysis, Validation, Investigation, Visualization, Methodology, Writing - original draft, Writing - review and editing; Anita X Sumali, Formal analysis; Tanmay P. Lele, Conceptualization, Formal analysis, Writing - original draft; Pushkar P. Lele, Conceptualization, Resources, Data curation, Software, Formal analysis, Supervision, Funding acquisition, Validation, Visualization, Methodology, Writing - original draft, Project administration, Writing - review and editing

### Author ORCIDs

Jyot D Antani (ID) https://orcid.org/0000-0002-7402-983X
Pushkar P. Lele (ID) https://orcid.org/0000-0002-2894-3487

### Decision letter and Author response

Decision letter https://doi.org/10.7554/eLife.63936.sa1
Author response https://doi.org/10.7554/eLife.63936.sa2

## Additional files

### Supplementary files

• Transparent reporting form

### Data availability

All data generated or analysed during this study are included in the manuscript and supporting files. Source data files have been provided for Figures 1-5 as Excel spreadsheets. References for the data in Figure 6 have been included in the caption.

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

## Appendix 1

### Δ*cheY* cells swim in the pusher mode only

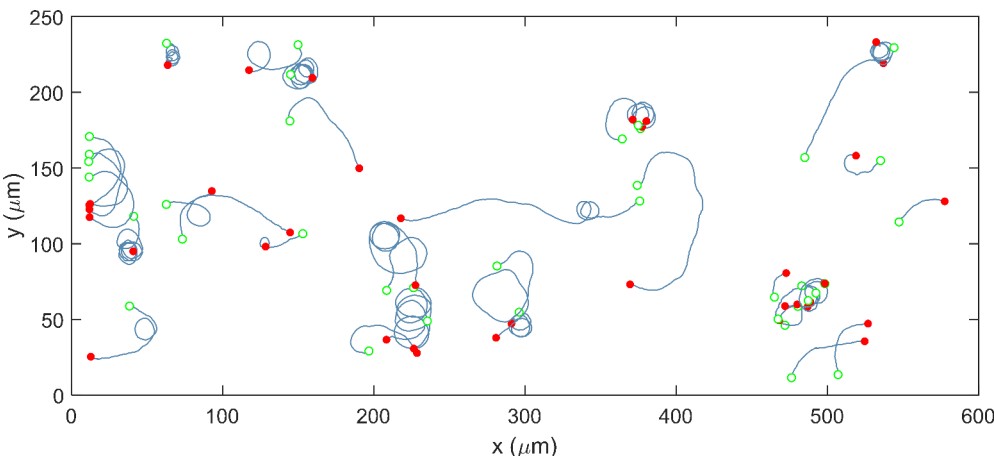

**Appendix 1—figure 1.** Single-cell trajectories of a *H. pylori* PMSS1 ΔcheY mutant are indicated. The cells swam in clockwise-only trajectories as shown: open green circles denote the start of a trajectory; filled red circles denote the end. This behavior was observed for n > 150 cells; here we show 38 cells.

## Appendix 2

### Estimation of dissociation constants

The rate constants for flagellar switching from CCW to CW ($k_{CCW \to CW}$) and CW to CCW ($k_{CW \to CCW}$) were estimated from the $CW_{bias}$ and the reversal frequencies, $\omega$:

$$k_{CCW \to CW} = \frac{\omega}{2(1 - CW_{bias})}$$

$$k_{CW \to CCW} = \frac{\omega}{2CW_{bias}}$$

As shown in **Appendix 2—figure 1**, both rate constants increased until the physiological temperature (37°C) was attained; the rates decreased thereafter. Reliable estimates of the rate constants could not be obtained at 25°C, owing to the low frequency of reversals at that temperature.

Following the model of **Scharf et al., 1998**, the ratio of the dissociation constant ($K$) to the concentration of the phosphorylated CheY ($C$) was calculated from:

$$\ln\left(K_{eq}\right) = \frac{-\Delta G^0}{kT} + m.\ln(K_{CCW}/K_{CW})\frac{C}{C + K}$$

where $K_{eq} = \frac{CW_{bias}}{1 - CW_{bias}}$, and $K$ is the weighted average of the dissociation constants, $K_{CCW}$ and $K_{CW}$, for CheY-P binding to the CCW and CW conformations, respectively. We assumed $K_{CCW}/K_{CW}$ = 4.7 from an earlier work in *E. coli* (**Fukuoka et al., 2014**). The standard free energy difference between the CW and CCW conformations in the absence of CheY-P, $\Delta G^0$, was estimated at the temperatures used in our work by extrapolating previous data (**Turner et al., 1996**). The number of CheY-P binding sites in the flagellar switch in *H. pylori*, was determined from the ratio of the sizes of the switch complexes in the two species (**Qin et al., 2017**): $m_{H.\,pylori} = m_{E.\,coli}\left(\frac{Diameter_{E.\,coli\,switch}}{Diameter_{H.\,pylori\,switch}}\right)$ where $m_{E.\,coli} = 34$ (**Lele et al., 2012**).

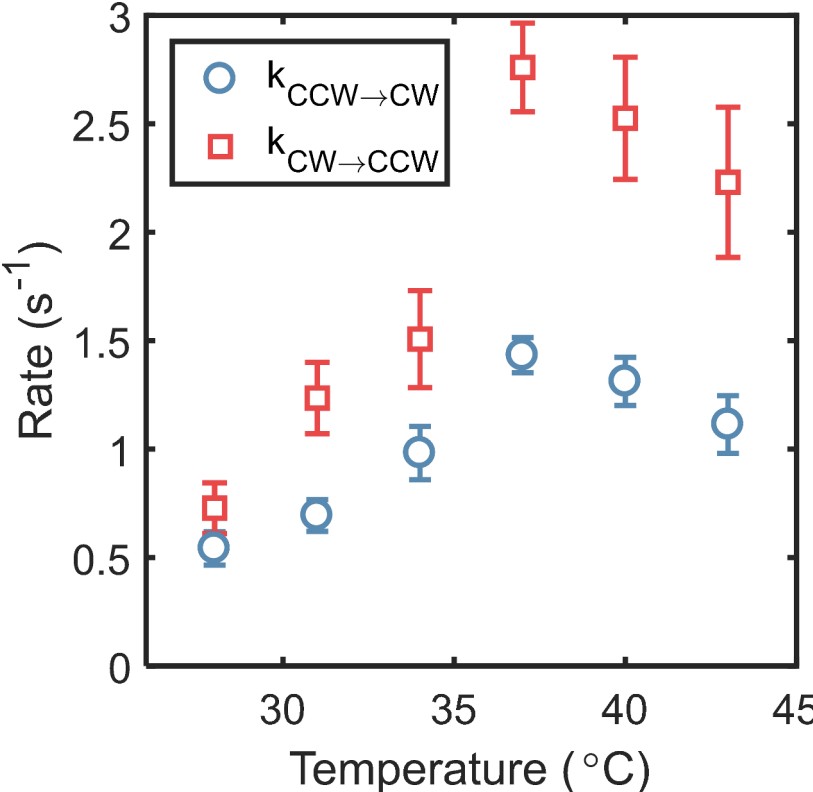

**Appendix 2—figure 1.** The switching rates were estimated from the $CW_{bias}$ and the reversal frequencies reported in **Figure 4B** (main text). The maximum $k_{CW \to CCW}$ and $k_{CCW \to CW}$ values were attained at 37°C (2.75 $\pm$ 0.20 s$^{-1}$ and 1.43 $\pm$ 0.08 s$^{-1}$). The standard error is indicated.

## Appendix 3

### Wait-times for pusher and puller modes

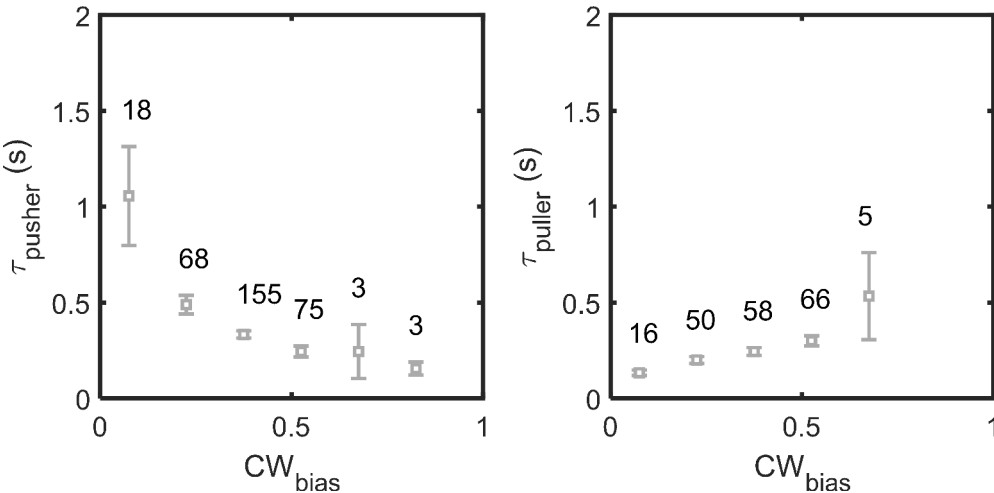

**Appendix 3—figure 1.** Variation in wait-times in the pusher and puller modes with $CW_{bias}$. Each point on the plot is calculated by averaging over the number of samples noted above the point.

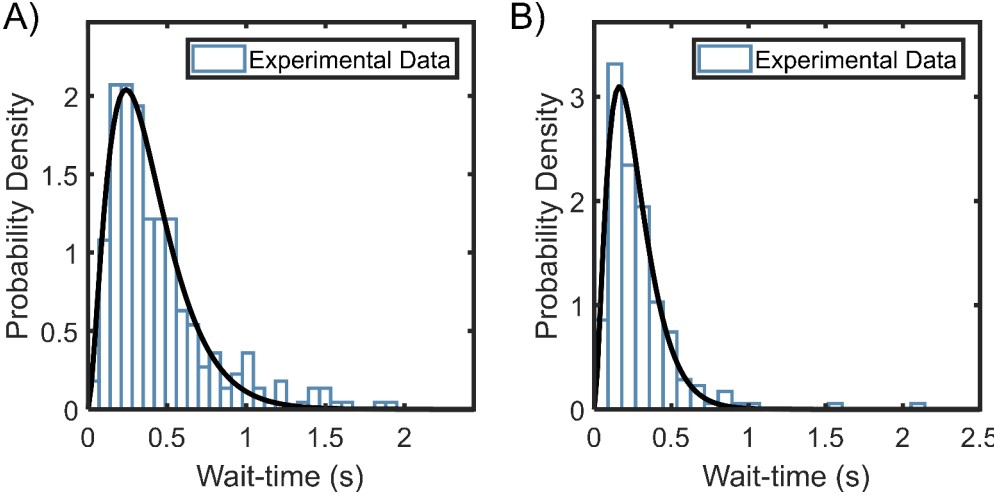

**Appendix 3—figure 2.** Wait-time distributions for (**A**) pusher mode (n = 322 segments) and (**B**) puller mode (n = 196 segments). Gamma-fits reveal that the mean ± variance in wait-times for the pusher mode is 0.38 ± 0.06 and that for the puller mode is 0.26 ± 0.02.

## Appendix 4

### Diffusion model for asymmetric swimmers

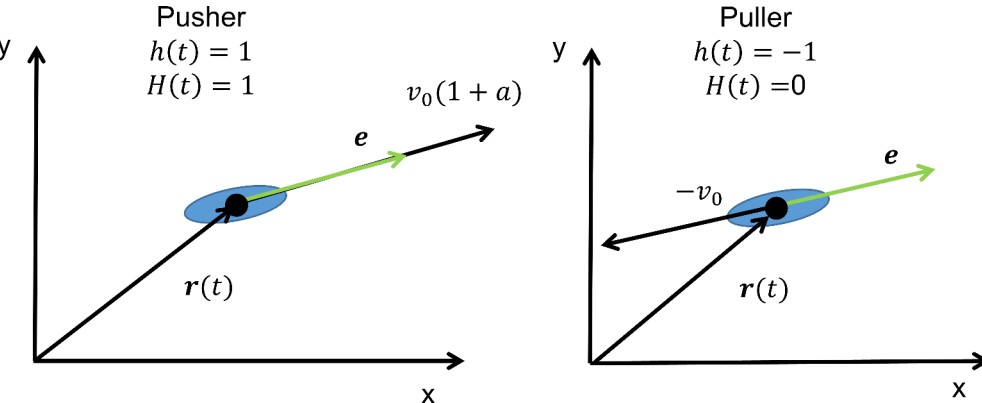

**Appendix 4—figure 1.** Cell alignment and position are defined by the vectors e and r. Function $h(t)$ alternates between $-1$ and 1 with each reversal. The Heaviside function, $H(h)$, describes the magnitudes of the two swimming modes with the asymmetry parameter $a$.

The velocity of the bacterium is expressed as:

$$\frac{d\boldsymbol{r}(t)}{dt} = \boldsymbol{v}(t) = v_0\, \boldsymbol{e}(t)\, h(t)\, [1 + aH(t)] \tag{1}$$

where the unit vector $\boldsymbol{e}(t)$ represents cell alignment. The state function $h(t)$ describes the direction of swimming, and alternates between $h(t) = 1$ and $h(t) = -1$ to indicate the two modes (**Appendix 4—figure 1**).

$$h(t) = \begin{cases} 1, & pusher \\ -1, & puller \end{cases}$$

A Heaviside function $H(t)$ characterizes the difference in swimming speeds in the two modes:

$$H(t) = \begin{cases} 1, & h(t)=1 \\ 0, & h(t)=-1 \end{cases}$$

Thus, the magnitudes of the two speeds are, $v_{pusher} = (1+a)v_0$ and $v_{puller} = v_0$, where $a$ is the asymmetry parameter.

Rotational diffusion causes the cell to deviate from a straight line during a run (or reversal), described by $\frac{d\theta}{dt} = \sqrt{2D_\theta}\xi(t)$, where the white noise is characterized by $\langle \xi(t) \rangle = 0$, and $\langle \xi(t)\xi(t+\tau) \rangle = \delta(-\tau)$. $D_\theta$ is the rotational diffusion coefficient.

The velocity autocorrelation is:

$$\langle \boldsymbol{v}(t)\boldsymbol{v}(t+\tau) \rangle = v_0^2 \langle \boldsymbol{e}(t)\boldsymbol{e}(t+\tau) \rangle \langle G(t,\tau) \rangle \tag{2}$$

here,

$$G(t,\tau) = h(t)h(t+\tau)[1+aH(t)][1+aH(t+\tau)] \tag{3}$$

Also, $\langle \boldsymbol{e}(t)\cdot\boldsymbol{e}(t+\tau) \rangle = e^{-D_\theta \tau}$ (**Mikhailov and Meinköhn, 1997**; **Schienbein and Gruler, 1993**).

The value of $G$ in time $\tau$ is influenced by the number of reversals, $k$, whether they are odd or even, and the initial mode of swimming at $t$ (see **Appendix 4—table 1**).

**Appendix 4—table 1.** Value of $G(t,\tau)$ for different possibilities.

For odd or even number of reversals occurring between time $t$ and $t+\tau$, corresponding cases of initial and final values for state function $h$ and Heaviside function $H$ are considered. Substituting $h$ and $H$ values in (3), $G$ is calculated for each case.

| Number of reversals | $h(t)$ | $h(t+\tau)$ | $H(t)$ | $H(t+\tau)$ | $G(t,\tau)$ |
|---|---|---|---|---|---|
| Even | 1 | 1 | 1 | 1 | $(1+a)^2$ |
| Even | −1 | −1 | 0 | 0 | 1 |
| Odd | 1 | −1 | 1 | 0 | $-(1+a)$ |
| Odd | −1 | 1 | 0 | 1 | $-(1+a)$ |

Based on table 2, the average value of the correlation can be determined from the probabilities of $k$ reversals, $P_k^{h(t),h(t+\tau)}(t,\tau)$.

$$\langle G(t,\tau)\rangle = (1+a)^2 P_0^{1,1}(t,\tau) + 1 \cdot P_0^{-1,-1}(t,\tau)$$
$$+\left[(1+a)^2 \cdot P_{even}^{1,1}(t,\tau) + 1 \cdot P_{even}^{-1,-1}(t,\tau)\right] \tag{4}$$
$$-(1+a)\left[P_{odd}^{1,-1}(t,\tau) + P_{odd}^{-1,1}(t,\tau)\right]$$

Assuming that the probability of finding the cell in the two modes initially (at time $t$) is similar ($CW_{bias} \sim 0.5$), the expression reduces to:

$$\langle G(t,\tau)\rangle = \frac{(1+a)^2+1}{2}\cdot(P_0(t,\tau)+P_{even}(t,\tau))-(1+a)\cdot P_{odd}(t,\tau) \tag{5}$$

The probability $P_{even}$ represents the cumulative probability that the cell undergoes an even and non-zero number of reversals ($k = 2, 4, 6,\ldots$). Similarly, the probability $P_{odd}$ represents the cumulative probability that the cell undergoes an odd number of reversals ($k = 1, 3, 5,\ldots$).

To determine the probabilities, we extended an approach previously developed by Großmann and co-workers for the case of an symmetric swimmer that stochastically reverses its direction of swimming (*Großmann et al., 2016*). Our experimental measurements suggested that the run-times obeyed a Gamma distribution (*Figure 5A*, main text):

$$\Omega(t) = \frac{r^M t^{M-1} e^{-rt}}{(M-1)!}$$

where, $M$ is the shape-parameter and $1/r$ is the scale-parameter. The probability of $k = 0$ reversals was then determined in the Laplacian space as (*Großmann et al., 2016*):

$$P_0(t,\tau) \leftrightarrow \hat{P}_0(s,u) = \frac{1}{u}\cdot\left[\frac{1}{s}-\frac{1}{1-\hat{\Omega}(s)}\cdot\frac{\hat{\Omega}(s)-\hat{\Omega}(u)}{u-s}\right] \tag{6}$$

where,

$$\Omega(s) = \left(\frac{r}{r+s}\right)^2 \tag{7}$$

To determine $P_{even}$, a summation of the even probabilities ($k = 2, 4, 6, \ldots$) was obtained while accounting for the turning angle, $\varnothing$:

$$P_{even}(t,\tau) \leftrightarrow \hat{\beta}_{even}(s,u) = \sum_{k=2,4,6,\ldots}^{\infty} \frac{1-\hat{\Omega}(u)}{u}\cdot\frac{\left(\hat{\Omega}(u)\right)^{k-1}}{1-\hat{\Omega}(s)}\cdot\frac{\hat{\Omega}(s)-\hat{\Omega}(u)}{u-s}\cdot\alpha^k$$

$$= \alpha\cdot\Upsilon(s,u)\sum_{k=2,4,6,\ldots}^{\infty}\left(\alpha\cdot\hat{\Omega}(u)\right)^{k-1} \tag{8}$$

$$\Upsilon(s,u) = \frac{1 - \hat{\Omega}(u)}{u} \cdot \frac{1}{1 - \hat{\Omega}(s)} \cdot \frac{\hat{\Omega}(s) - \hat{\Omega}(u)}{u - s} \tag{9}$$

The series summation reduces to:

$$\sum_{k=2,4,6,\ldots}^{\infty} \left(\alpha \cdot \hat{\Omega}(u)\right)^{k-1} = \frac{\alpha \cdot \hat{\Omega}(u)}{1 - \left(\alpha \cdot \hat{\Omega}(u)\right)^2} \tag{10}$$

Here, $\alpha = |\langle cos\emptyset \rangle|$. The turning angle, $\emptyset$, randomizes the bacterial trajectory similar to rotational diffusion but only acts upon a reversal.
Similarly,

$$P_{odd}(t,\tau) \leftrightarrow \hat{\beta}_{odd}(s,u) = \alpha \cdot \Upsilon(s,u) \sum_{k=1,3,5,\ldots}^{\infty} \left(\alpha \cdot \hat{\Omega}(u)\right)^{k-1} \tag{11}$$

The series summation reduces to:

$$\sum_{k=1,3,5,\ldots}^{\infty} \left(\alpha \cdot \hat{\Omega}(u)\right)^{k-1} = \frac{1}{1 - \left(\alpha \cdot \hat{\Omega}(u)\right)^2} \tag{12}$$

To estimate the long-time probabilities, the final value theorem was employed:

$$\hat{P}_k(t \to \infty, \tau) = \lim_{s \to 0} s \cdot \hat{P}_k(s,u) \tag{13}$$

Substituting (*Equation 6*) and (*Equation 7*) in (*Equation 13*),

$$P_0(t \to \infty, \tau) = \frac{1}{u} - \frac{r}{u^2 M}\left(1 - \hat{\Omega}(u)\right) \tag{14}$$

Substituting (*Equation 7, 8 and 9*), and (*Equation 10*) in (*Equation 13*),

$$P_{even}(t \to \infty, \tau) \leftrightarrow \lim_{s \to 0} s \cdot \hat{\beta}_{even}(s,u) = \frac{r}{M} \cdot \frac{\hat{\Omega}(u)}{u^2} \cdot \frac{\alpha^2 \left(1 - \hat{\Omega}(u)\right)^2}{1 - \alpha^2 \cdot \hat{\Omega}(u)^2} \tag{15}$$

Similarly,

$$P_{odd}(t \to \infty, \tau) \leftrightarrow \lim_{s \to 0} s \cdot \hat{\beta}_{odd}(s,u) = \frac{r}{M} \cdot \frac{1}{u^2} \cdot \frac{\alpha \left(1 - \hat{\Omega}(u)\right)^2}{1 - \alpha^2 \cdot \hat{\Omega}(u)^2} \tag{16}$$

Combining *Equations (14), (15), (16), and (5)*,

$$\hat{G}(u) = \frac{(1+a)^2 + 1}{2} \cdot \left\{ \frac{1}{u} - \frac{r}{u^2 M}\left(1 - \hat{\Omega}(u)\right) + \frac{r}{M} \cdot \frac{\hat{\Omega}(u)}{u^2} \cdot \frac{\alpha^2 \left(1 - \hat{\Omega}(u)\right)^2}{1 - \alpha^2 \cdot \hat{\Omega}(u)^2} \right\}$$
$$- (1+a) \cdot \left\{ \frac{r}{M} \cdot \frac{1}{u^2} \cdot \frac{\alpha \left(1 - \hat{\Omega}(u)\right)^2}{1 - \alpha^2 \cdot \hat{\Omega}(u)^2} \right\} \tag{17}$$

For two-dimensions, the diffusion coefficient $D$ is related to the average correlation over long-times as (*Großmann et al., 2016*):

$$D = \frac{v_0^2}{2} G(D_\theta)$$

Finally, we obtain the following expression for the diffusion coefficient:
or,

$$D = \frac{v_0^2}{2D_\theta} \left[ \frac{(1+a)^2+1}{2} \cdot \left\{ \begin{array}{l} 1 - \frac{r}{MD_\theta}\left(1 - \frac{r^M}{(r+D_\theta)^M}\right) \\[2mm] + \frac{\alpha^2 r}{MD_\theta} \cdot \frac{r^M}{(r+D_\theta)^M} \cdot \frac{\left((r+D_\theta)^M - r^M\right)^2}{(r+D_\theta)^{2M} - \alpha^2 r^{2M}} \end{array} \right\} \\[4mm] -(1+a)\left\{ \frac{\alpha \cdot r}{MD_\theta} \cdot \frac{\left((r+D_\theta)^M - r^M\right)^2}{(r+D_\theta)^{2M} - \alpha^2 r^{2M}} \right\} \right]$$

(18)

For $a = 0$ (no asymmetry in speeds) and $\alpha = 1$ (180° reversals), equation 3.13 from *Großmann et al., 2016* is recovered.

A Gamma fit to the experimentally-determined wait-time distributions yielded M = 3 (*Figure 5A*, main text), such that:

$$D = \frac{v_0^2}{2D_\theta} \left[ \frac{(1+a)^2+1}{2} \cdot \left\{ \begin{array}{l} 1 - \frac{\omega}{D_\theta}\left(1 - \frac{(3\omega)^3}{(3\omega+D_\theta)^3}\right) \\[2mm] + \frac{|\langle cos\phi\rangle|^2 \omega}{D_\theta} \cdot \frac{(3\omega)^3}{(3\omega+D_\theta)^3} \cdot \frac{\left((3\omega+D_\theta)^3 - (3\omega)^3\right)^2}{(3\omega+D_\theta)^6 - |\langle cos\phi\rangle|^2(3\omega)^6} \end{array} \right\} \\[4mm] -(1+a)\left\{ \frac{|\langle cos\phi\rangle| \cdot \omega}{D_\theta} \cdot \frac{\left((3\omega+D_\theta)^3 - (3\omega)^3\right)^2}{(3\omega+D_\theta)^6 - |\langle cos\phi\rangle|^2(3\omega)^6} \right\} \right]$$

(19)

Here, the reversal frequency $\omega = r/M$.

## Appendix 5

### Single-cell simulation

Each cell was initialized at the origin ($x = 0, y = 0$) at time $t = 0$ s. Cell movement was simulated as alternating runs and reversals over a total duration of ~ 350 s. At the beginning of each run (or reversal), the time interval $\tau_i$ for the run (or reversal) was sampled from a Gamma distribution that was generated based on fits to experimental measurements (*Figure 5A*, main text). Each i$^{th}$ time interval $\tau_i$ was then divided into $n$ batches of equal durations, $\Delta t$. The time-step $\Delta t$ was fixed at 10 ms. Any remainder, $rem(\tau_i, \Delta t) = \zeta$ ($<\Delta t$), was allocated to an $(n+1)^{th}$ batch. Within each batch, the cell was assumed to travel in a straight line with a displacement given by:

$\varepsilon_n = v_n \zeta$n+1, where $\zeta = \Delta t$n+1 for each of the $n$ bins and $\zeta = \tau_i - n\Delta t$n+1 for the final $n+1^{th}$ bin.

$$\therefore \varepsilon_n = h(t + \tau_i)v_o(1 + aH(t + \tau_i))\boldsymbol{e}_n \times \zeta \qquad (20)$$

Here, $h$ is either +1 or −1 for a given interval:

$$h(t + \tau_i) = -h(t + \tau_{i+1}) \text{ and } |h(t + \tau_i)| = 1 \qquad (21)$$

The Heaviside function: $H = 1 \rightarrow h = 1$ and $H = 0 \rightarrow h = -1$. The position vector is simply: $\boldsymbol{e}_n = \cos(\theta_n)\boldsymbol{\delta_i} + \sin(\theta_n)\boldsymbol{\delta_j}$

The angle $\theta_n$ was updated in between batches to account for rotational Brownian motion:

$$\theta_{n+1}^i = \theta_n^i + (2D_\theta\zeta)^{0.5} \qquad (22)$$

The $x$ and $y$ positions over time were calculated from, $x_k = \sum_1^k \delta_i.\varepsilon_n$, $y_k = \sum_1^k \delta_j.\varepsilon_n$. A sample trajectory for one such interval consisting of $n+1 = 71$ batches is indicated in *Appendix 5—figure 1A*.

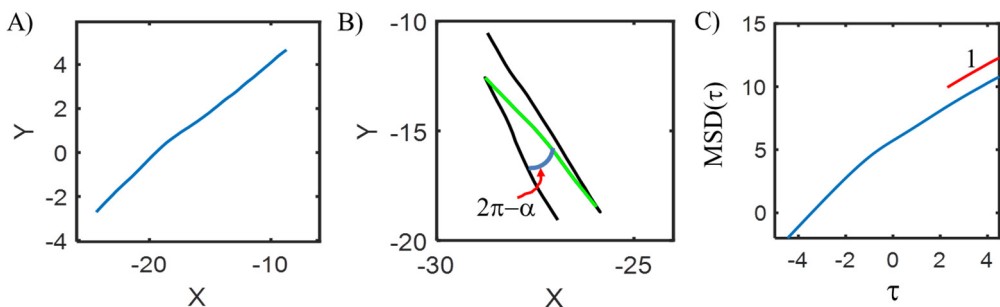

**Appendix 5—figure 1.** (**A**) A representative cell trajectory as it engages in a simulated run. The total run interval, $\tau$, was split into 71 batches. The trajectory is not a straight-line owing to Brownian motion. (**B**) Two simulated reversals are indicated. Black and green segments indicate pusher ($v = v_o(1 + a)$) and puller ($v = v_o$) modes, respectively. (**C**) Bacterial movements became purely diffusive over long times (~ 100 s) as indicated by the linear dependence of MSD on lag time $\tau$ (log-log plot). The value of the diffusion coefficient was calculated at these long times.

The turn angle, ∅, is the angle between the original direction just before and the new direction just after a reversal. The angle tends to randomize the bacterial random walk, in addition to Brownian motion. The turn angle is likely influenced by kinematic properties of the cell body and filaments, as well as the flexibility of the flagellar hook. To account for the turning angle, at the start of each time interval $\tau_i$, the angle $\theta$ was updated as:

$$\theta_1^{i+1} = \theta_{end}^i + (2D_\theta\zeta)^{0.5} + \emptyset^{i+1}$$

Note that this update only occurred at the start of each interval; subsequent batch-updates for $\theta$ within the interval occurred as per equation 22. The angles $\emptyset^{i+1}$ were sampled from a distribution that was obtained from fits to the experimental data (*Figure 1B*, main text). A representative reversal with the turning angle is shown in *Appendix 5—figure 1B*.

## Diffusion coefficients

The simulations were repeated for 1000 cells. Mean square displacements were calculated as:

$$MSD(\tau) = \left\langle (x(t+\tau) - x(t))^2 + (y(t+\tau) - y(t))^2 \right\rangle$$

The MSD versus the lag time $\tau$ became linear at long times (~100 s), indicating purely diffusive behavior (*Appendix 5—figure 1C*). The diffusion coefficient was calculated from the slope of MSD versus $\tau$ over these times (D = slope/4 for two-dimensional walk).

## Simulated diffusion with varying asymmetry and CW$_{bias}$

Following the scheme described above, we simulated movements of cells over varying CW$_{bias}$. To vary the bias, the mean wait times in the pusher and puller modes were varied. The wait times were Gamma distributed and the sum of the intervals ($\tau_{i,pusher}$ and $\tau_{i,puller}$) was fixed at 0.35 s. Diffusion coefficients were calculated by simulating 1000 cells each for the conditions $a$ = 0, 0.25, 0.5, 0.75, and 1; for CW$_{bias}$ = 0.14, 0.25, 0.39, 0.48, 0.61, 0.75, and 0.86. Results are plotted in *Figure 5D* (main text).

# Appendix 6

## Swimming asymmetry away from bounding surfaces

Differences in swimming speeds in the pusher and puller modes have been reported for *Vibrio alginolyticus*, but only near surfaces (*Magariyama et al., 2005*). A similar asymmetry was observed in *Pseudomonas putida* in the bulk fluid (*Theves et al., 2013*). To determine if near-wall effects played a role in the speed asymmetry in *H. pylori*, we recorded motile cells in the bulk, away from surfaces. To determine if near-wall hydrodynamic effects influence the asymmetry in *H. pylori*, we focused the microscope objective ~200 μm in the bulk fluid and recorded motility away from any surfaces in a culture-dish (see Materials and methods). *Appendix 6—figure 1* shows one such trajectory. We observed that swimming speeds in consecutive segments were anti-correlated, in a similar manner as trajectories near surfaces (refer to *Figure 1* in the main text of the manuscript). Although this observation has been made for a small number of cells (n = 4 cells), this preliminary data suggests that the asymmetry is unlikely to be due to the presence of nearby surfaces.

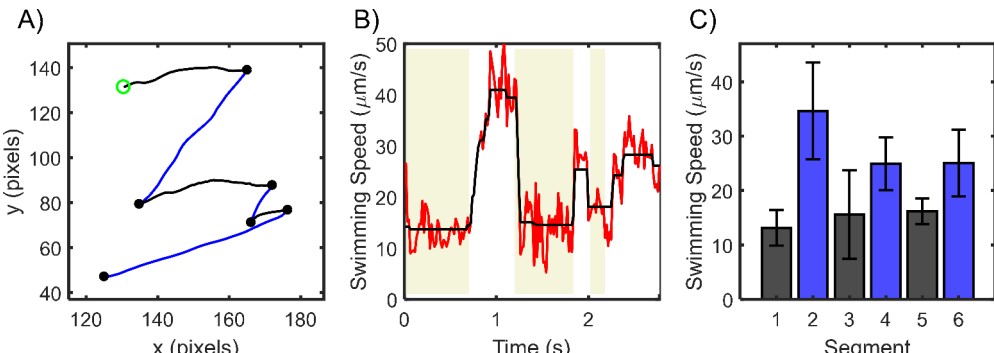

**Appendix 6—figure 1.** Cells of *H. pylori* exhibit asymmetry ~200 μm away from surfaces. (**A**) Trajectory of a representative cell is shown, where the segment-color changes upon each reversal. Beginning of the trajectory is denoted by a green circle, reversals are denoted by black circles. (**B**) Quantitatively determined speeds of the same cell. The shaded regions indicate alternating swimming modes. (**C**) Average speed for each segment along with the standard deviation is indicated in a chronological manner.

