## [Decision Letter]

**Acceptance summary:**

In this work, the authors measure the trajectory of hundreds of swimming H. pylori cells near a surface. Taking advantage of the hydrodynamic interactions of rotating flagella with a surface, they can identify the pusher and puller mode of locomotion and quantify the velocity in these two distinct modes of locomotion. The authors find that cells decrease the rotational bias in the presence of an attractant, similar to what observed in *E. coli*. While the reversal frequency has received much attention in H. pylori, these results demonstrate that rotational bias is critical to quantify the chemotactic output.

**Decision letter after peer review:**

[Editors’ note: the authors submitted for reconsideration following the decision after peer review. What follows is the decision letter after the first round of review.]

Thank you for submitting your work entitled "Anisotropic random walks reveal chemotaxis signaling output in run-reversing bacteria" for consideration by *eLife*. Your article has been reviewed by three peer reviewers, and the evaluation has been overseen by a Reviewing Editor and a Senior Editor. The reviewers have opted to remain anonymous.

Our decision has been reached after consultation between the reviewers. Based on these discussions and the individual reviews below, we regret to inform you that your work will not be considered further for publication in *eLife*.

The discussion reached the shared conclusion that the present work does not provide sufficiently novel insight to grant publication in *eLife*. The interesting technique to disentangle CW from CCW modes by taking advantage of interaction with the surface has been previously used (Magariyama et al., 2005; Raatz et al. Eur. Phys. J. 2015); previous work already showed that H. pylori swims faster in forward mode rather than in backward mode (Constantino et al., 2016); the steady state bias has been previously quantified with a different technique (Howitt et al., 2011). To enrich the results and provide biological insight for this problem the author may consider to quantify thoroughly how the bias is affected by chemoattractants and chemorepellents, in which case we would consider the manuscript as a new submission.

Reviewer #1:

Antani et al. describe precise speed measurements of Helicobacter pylori, a bacterium that is used as a model of chemotaxis and motility in mammalian hosts. A main finding is that H. pylori swim with a faster speed in pusher mode, and a slower speed in puller mode. The finding that H. pylori can swim in a pusher or puller mode was already reported and appropriately referenced, but the new part here is the speed differences and use of them. Modeling suggests this anisotropic speed behavior limits diffusive behavior at physiological temps, a very interesting finding. Overall, the paper is an elegant combination of careful microscopy and modeling to understand interesting microbial behavior.

Reviewer #2:

The paper "Anisotropic random walks reveal chemotaxis signaling output in run-reversing bacteria" is primarily an experimental study of the locomotion of the bacterium H. pylori. I should preface by saying that I work in biophysics/biomechanics but I am not a biologist.

The main idea in this paper is that it is difficult to perform the standard cell-tethering experiments done with *E. coli* to measure bias in the motor rotation because H. pylori is a polar lophotrichous cell with all flagellar filaments sticking out from one side of the cell. The authors measure in swimming *H. pylori* cells two clearly distinct modes of swimming, but it is not clear which one corresponds to CCW vs. CW rotation of the motors. So they had the idea, which I find clever, to use the fact that we know that different flagellar rotation directions leads to circular tracks with different directions near surfaces. Doing this, they can identify which mode is which: the CCW motor rotation (pusher mode) is about 1.5x faster than the CW motor rotation (puller mode).

It is a very clear and well-written paper; it focuses on a single issue and treats it very convincingly. The section on the effect of temperature is a bit more foreign to me (very biological) but the rest was both interesting and appears to be correct. I am very happy to recommend this for publication.

Reviewer #3:

Measurements of the swimming behavior of H. pylori and the following claims are presented:

1) H. pylori swims faster with the flagellated pole lagging than leading.

2) The rotational bias (fraction of time spent in one flagellar rotation state) can be extracted from the individual distribution of swimming speeds.

3) The bias in rotation direction reports on "chemotactic output".

4) Different forward and backward running speeds increase diffusive spreading in run-reverse motility.

5) Swimming speed and turning frequency vary with temperature.

Even if all claims were novel and supported, the resulting impact would not justify publication in e*Life*. However, only claim 2 and 5 are. Some of the findings have descriptive value of interest to a section of the microbiology community.

Substantive concerns

1) The H. pylori literature background and the context of the present work are insufficiently discussed, making it laborious to assess the novelty of the claims.

2) The central claim (3) that bias reports on "chemotactic output" rests on an assumption that is not supported and that seems less likely to be true than not, namely that *H. pylori* performs chemotaxis by modifying the bias in flagellar rotation direction like *E. coli*.

a) Many other polarly flagellated species chemotact by changing turning the frequency without substantial change in bias: *V. alginolyticus* (Xie, Lu, Wu. Biophys. J. 2015), *P. aeruginosa*: (Cai et al., mbio 2016), *C. crescentus* (Grognot et al., bioRxiv 2020).

b) *H. pylori's* polar flagella enable locomotion with either direction of rotation, whereas *E. coli*'s peritrichous flagella only enable locomotion in one rotation state. Thus bias modification makes sense for *E. coli*, but not for *H. pylori*.

c) The existing H. pylori literature also focusses on the turning frequency.

d) To support their assumption, the authors refer to the fact that modifying the turning frequency by concentration does not yield chemotaxis. That is a logical fallacy – the point made in Appendix C of the cited source (Berg's "Random Walks in Biology") is that chemotaxis is achieved by modifying turning frequency in response to changes in chemical concentration, rather than absolute concentrations. It is not a statement on the effectiveness of bias modulation vs turning frequency modulation. If the authors are referring to a different section of the book, they should state explicitly which one.

e) The "similarity in rotational bias in the two species" is referred to as grounds for assuming similarity in chemotactic strategy. Not only is the bias not similar (*H. pylori* shows similar durations of CCW and CW intervals, while in *E. coli* the typical wildtype CW bias is around ~10-15% (Montrone et al., 1998, Liu et al., 2020 and many others – though Ford et al., 2018 from the authors' lab reports a higher value), the argument also does not hold.

f) Given the crucial importance of the assumption, solid experimental evidence for it should be presented. That could be e.g. data that show that the bias changes drastically when an attractant/repellent is added.

3) Claim 1 is not novel (see e.g. Constantino *et al*., Science Advances 2016).

4) Claim 4 is not novel (see Theves *et al*., 2013).

5) The Discussion contains a number of misrepresentations of the literature.

[Editors’ note: further revisions were suggested prior to acceptance, as described below.]

Thank you for submitting your article "Asymmetric random walks reveal that the chemotaxis network modulates flagellar rotational bias in Helicobacter pylori" for consideration by *eLife*. Your article has been reviewed by two peer reviewers, and the evaluation has been overseen by a Reviewing Editor and Aleksandra Walczak as the Senior Editor. The following individual involved in review of your submission has agreed to reveal their identity: Christian Esparza-Lopez (Reviewer #1).

The reviewers have discussed the reviews with one another and the Reviewing Editor has drafted this decision to help you prepare a revised submission.

Summary:

In this work, the authors measure the trajectory of hundreds of swimming H. pylori cells near a surface. Taking advantage of the hydrodynamic interactions of rotating flagella with a surface, they can identify the pusher and puller mode of locomotion and quantify the velocity in these two distinct modes of locomotion. The authors find that cells decrease the rotational bias in the presence of an attractant, similar to what observed in *E. coli*. While the reversal frequency has received much attention in H. pylori, these results demonstrate that rotational bias is critical to quantify the chemotactic output.

Revisions:

1) the authors state "a very low basal value of the CWbias is disadvantageous as it prevents a response to an attractant stimulus – the cells cannot respond to an attractant if the pre-stimulus bias is ~ 0.", and similarly "But, a very low basal value of the CWbias is disadvantageous as it prevents cells from responding to attractants.". Is this a well-known experimental fact? In which case it may be useful to add a citation. Or do the authors infer this from their own data? In which case I missed something.

2) I think it would be helpful to discuss in the main text the results shown in Figure 5D and E. In particular, for a given asymmetry, the coefficient of diffusion first goes down with CWbias then goes up again, so that for small asymmetries the coefficient of diffusion is similar at low (~0.1) and high (~0.9) CWbias. I guess the diffusion coefficient first goes down because the fraction of time spent in the slow mode increases, but then why does it go up again?

3) As I said above, I like the data shown in Figure 4C, which demonstrates that the reversal frequency is likely not a good descriptor for chemotaxis. From what I understand, we're in fact hiding two reversal frequencies in the reversal frequency: the one from puller to pusher, and the one from pusher to puller. As for the CWbias, it tells us about the fraction of time in CW swimming, but not about the time spent in CW swimming. So I'm wondering if there's not a time information missing in the CWbias. Have the authors tried to look at the two reversal frequencies separately? Maybe the frequency of switching from puller to pusher is a more complete description of the data? Or have the authors plotted the CWbias on the x axis, and on the y axis the reversal frequency of slow to fast mode, and the reversal frequency of fast to slow mode? Although I am not asking the authors to redo the analysis, I think if they already have looked at these ways of describing the data, it would be useful for the reader to know it, and potentially to have the corresponding graphs as supplementary material.

4) Figure 3C and 3D: what are the light gray bins? The legend only indicates white for control and dark gray for attractant.

---

## [Author Response]

[Editors’ note: the authors resubmitted a revised version of the paper for consideration. What follows is the authors’ response to the first round of review.]

The discussion reached the shared conclusion that the present work does not provide sufficiently novel insight to grant publication in eLife. The interesting technique to disentangle CW from CCW modes by taking advantage of interaction with the surface has been previously used (Magariyama et al., 2005; Raatz et al. Eur. Phys. J. 2015); previous work already showed that H. pylori swims faster in forward mode rather than in backward mode (Constantino et al. Science Advances 2016); the steady state bias has been previously quantified with a different technique (Howitt et al., 2011). To enrich the results and provide biological insight for this problem the author may consider to quantify thoroughly how the bias is affected by chemoattractants and chemorepellents, in which case we would consider the manuscript as a new submission.

In the previous version of the manuscript, we discussed an assay that enabled us to track the probability of clockwise rotation (rotational bias) in the flagellar motor in *H. pylori*. In the revised work, we have used that assay as suggested by the editor/reviewers to perform studies on how the bias is modulated by the chemotaxis network in response to a chemoeffector. We report the following novel findings:

1) We have shown that *H. pylori* decrease the rotational bias of flagellar motors (CW_bias_) in response to a potent chemo-attractant, urea (Figure 3C in the revised manuscript). This is similar to how *E. coli* modulates rotational bias in response to chemo-attractants.

2) We observe that the default direction of rotation in *H. pylori* is counterclockwise (CCW) in the absence of CheY, a molecule that links the chemotaxis network to the flagellar motors (Figure 3B). In the presence of CheY-P, the rotational bias increases. This is true of *E. coli* as well.

3) The relationship between CW_bias_ and the reversal frequency is unimodal (Figure 4C), similar to that observed in *E. coli*. This is the first systematic characterization of the dependence of reversal frequencies on the rotational bias in polar –flagellates to our knowledge

Significance and impact: The novel insights reported above suggest that the chemotaxis network in *H. pylori* modulates the flagellar rotational bias in a similar manner to *E. coli*. We have modified the title accordingly. These findings are significant for the following reasons:

1) Reviewer 3 had predicted that the chemotaxis network in *H. pylori* was not likely to modulate the rotational bias as *E. coli* does, as the former is a polar-flagellate whereas the latter is peritrichous. Reviewer 3 had predicted that *H. pylori* were likely to adopt a frequency-modulation strategy for chemotaxis, unlike *E. coli*. They also had predicted that the dependence of *H. pylori* reversal frequency on the bias would not be similar to *E. coli*.

Our new results directly contradict all these predictions (see detailed responses to reviewer 3). Based on the above results, it is evident that if *H. pylori* adopted the frequency-modulation strategy suggested by reviewer 3, the chemotaxis network would likely be rendered superfluous. Thus, our results challenge the current understanding and intuition regarding chemotaxis in polar-flagellates.

2) Reviewer 3 correctly pointed out that the *H. pylori* field focuses on reversal frequencies as measure of chemotaxis response. Our results explain why this focus is misleading and why the rotational bias is a better measure of the chemotaxis output. In the canonical model, the CW_bias_ is indicative of the chemotaxis kinase activity – when CW_bias_ is high, the activity is high and vice-versa (see Cluzel et al., 2000). The architecture of the chemotaxis network in *H. pylori* shares strong similarities with the canonical model- *E. coli* (Lertsethtakarn et al., 2011). As we have now shown, the flagellar switch-functions are also similar in the two species. As is clearly evident from Figure 4C, a decrease in reversal frequency can indicate that either the CW_bias_ (kinase activity) has increased or that it has decreased. But attractant decrease kinase activity; repellents increase kinase activity. Hence, we propose that reversal frequencies should not be used to report chemotaxis kinase responses to chemoeffectors. Yet, previous work in *H. pylori* has relied on changes in reversal frequencies alone to characterize chemotaxis responses (Schweinitzer et al., 2008; Rader et al., 2011; Sweeney et al., 2012; Sanders et al., 2013; Collins et al., 2016; Machuca et al., 2017). Therefore, we call for a change in current approaches for characterizing flagellar responses in polar flagellates based on this result.

3) Our findings (1, 2, 3) suggest that *H. pylori* modulate flagellar responses similar to *E. coli*. This is backed by known similarities in the flagellar motor structure and the makeup of the chemotaxis network in the two species. Thus, our work provides key functional information that should guide ongoing and future studies of the flagellar motor and the chemotaxis receptors/auxiliary enzymes.

4) Considering that polar-flagellates (*H. pylori*) undergo displacement in both modes unlike the peritrichous *E. coli*, our findings raise major questions: what exactly is the role of the chemotaxis kinase in *H. pylori* chemotaxis? Is the forward mode indeed more suitable for exploration in polar-flagellates, as has been suggested by Xie, Lu, and Wu, *Biophys J*, 2015 for *V. alginolyticus*, or is that true only in the case of certain species? Hence, our work will inspire significant research in the future on the chemotaxis mechanisms in *H. pylori* and other polar-flagellates in the future.

5) Reviewer 3 referred to an alternate technique employed in two previous works on the asymmetry in swimming in *H. pylori* (Constantino et al., 2016 and Howitt et al., 2011). Owing to the limitation of this technique, these studies based conclusions (specific to asymmetry and bias) on sample sizes consisting of 1 wild-type cell (Constantino et al., 2016) and 8 wild-type cells (Howitt et al., 2011). In contrast, our method allowed us to derive conclusions regarding flagellar functions from hundreds of wild-type cells (100 – 900 cells). We believe our work will be valuable to researchers because it raises the standard for scientific rigor for *H. pylori* studies.

Considering the biomedical relevance of *H. pylori*, which infect more than half the world population and cause stomach cancers, our findings on chemotaxis signaling will appeal the broad audience that *eLife* targets.

Reviewer #1:Antani et al. describe precise speed measurements of Helicobacter pylori, a bacterium that is used as a model of chemotaxis and motility in mammalian hosts. A main finding is that H. pylori swim with a faster speed in pusher mode, and a slower speed in puller mode. The finding that H. pylori can swim in a pusher or puller mode was already reported and appropriately referenced, but the new part here is the speed differences and use of them. Modeling suggests this anisotropic speed behavior limits diffusive behavior at physiological temps, a very interesting finding. Overall, the paper is an elegant combination of careful microscopy and modeling to understand interesting microbial behavior.

We thank the reviewer for the positive comments. We have clarified our main findings and its potential impact in the response above.

Reviewer #2:The paper "Anisotropic random walks reveal chemotaxis signaling output in run-reversing bacteria" is primarily an experimental study of the locomotion of the bacterium H. pylori. I should preface by saying that I work in biophysics/biomechanics but I am not a biologist.The main idea in this paper is that it is difficult to perform the standard cell-tethering experiments done with *E. coli* to measure bias in the motor rotation because H. pylori is a polar lophotrichous cell with all flagellar filaments sticking out from one side of the cell. The authors measure in swimming *H. pylori* cells two clearly distinct modes of swimming, but it is not clear which one corresponds to CCW vs. CW rotation of the motors. So they had the idea, which I find clever, to use the fact that we know that different flagellar rotation directions leads to circular tracks with different directions near surfaces. Doing this, they can identify which mode is which: the CCW motor rotation (pusher mode) is about 1.5x faster than the CW motor rotation (puller mode).It is a very clear and well-written paper; it focuses on a single issue and treats it very convincingly. The section on the effect of temperature is a bit more foreign to me (very biological) but the rest was both interesting and appears to be correct. I am very happy to recommend this for publication.

We thank the reviewer for these positive comments. We have now provided context to the temperature experiments; they enabled us to extract the dependence of reversal frequency on rotational bias, which challenges one of the major predictions by reviewer 3. These and other main advances from our new work and its impact has been discussed in the response above. Also see responses to reviewer 3.

Reviewer #3:Measurements of the swimming behavior of H. pylori and the following claims are presented:1) H. pylori swims faster with the flagellated pole lagging than leading.2) The rotational bias (fraction of time spent in one flagellar rotation state) can be extracted from the individual distribution of swimming speeds.3) The bias in rotation direction reports on "chemotactic output".4) Different forward and backward running speeds increase diffusive spreading in run-reverse motility.5) Swimming speed and turning frequency vary with temperature.

We have listed the key findings and the impact from the revised work in our response to this reviewer and editor’s comments. We have significantly updated the text to clarify the main advances from our work based on reviewer 3 and other reviewers’ comments. These comments have considerably improved the impact of the work (in our opinion). We are grateful, thank you!

Even if all claims were novel and supported, the resulting impact would not justify publication in eLife.

We observed that *H. pylori* modulate the rotational bias similar to *E. coli* in response to a strong attractant (see Figure 3C, D), where the gray data indicates response to attractant and the white data indicates a control. We also show that the bias in *H. pylori* is zero in the absence of CheY (Figure 3B) and that the presence of CheY increases the bias, similar to *E. coli* (Figure 3A).

Although reviewer 3 had predicted that the reversal frequency versus rotational bias relationship would not be similar between the two species, our data suggest otherwise (Figure 4C). These results highlight the inaccuracies prevalent in current approaches in the field, which rely on quantification of the reversal frequencies in *H. pylori* for characterizing chemotaxis response (Schweinitzer et al., 2008; Rader et al., 2011; Sweeney et al., 2012; Sanders et al., 2013; Collins et al., 2016; Machuca et al., 2017). Also see Discussion section.

Finally, our observations of functional similarities in the chemotaxis signaling between the two species is significant as polar-flagellates such as *H. pylori* have the ability to reverse their paths. To chemotax, they likely rely on strategies that are currently unknown. Our findings call for investigations into the mechanisms of chemotaxis in this particular species, and possibly in others (see Discussion section).

However, only claim 2 and 5 are. Some of the findings have descriptive value of interest to a section of the microbiology community.

We are glad that the reviewer agrees that our method to extract rotational bias is novel. We now show how valuable the temperature experiments will be for all scientific communities interested in chemotaxis – these experiments helped discover the dependence of reversal frequencies on the rotational bias in *H. pylori* for the first time in a polar-flagellate (to our knowledge).

Substantive concerns1) The H. pylori literature background and the context of the present work are insufficiently discussed, making it laborious to assess the novelty of the claims.

We believe that we had cited all the prominent literature relevant to chemotaxis in *H. pylori* including Constantino et al., 2016 and Howitt et al., 2011. We have now added several related works for chemotaxis in polar-flagellates. Specifically, we cite: Constantino et al., 2016; Howitt et al., 2011; Xie et al., 2015; Cai et al., 2015; and Morse et al., 2016. We would be happy to add other citations that we may have missed.

2) The central claim (3) that bias reports on "chemotactic output" rests on an assumption that is not supported and that seems less likely to be true than not, namely that *H. pylori* performs chemotaxis by modifying the bias in flagellar rotation direction like *E. coli*.

Our new experiments argue against this notion that *H. pylori* do not modify the flagellar bias similar to *E. coli*. We showed that:

1) In the absence of CheY, motors rotate CCW-only.

2) In the presence of CheY, the bias is higher (~ 0.35).

3) Treatment with attractant, which is expected to decrease the kinase activity, reduces the bias. The decrease in reversal frequency occurs as a consequence.

4) And the reversal frequency unimodally depends on the bias.

These are key similarities between *E. coli* and *H. pylori*. Our functional data are backed by the structural similarity in the motor and the architectural similarity in the chemotaxis network (see Discussion section for references).

a) Many other polarly flagellated species chemotact by changing turning the frequency without substantial change in bias: *V. alginolyticus* (Xie, Lu, Wu. Biophys. J. 2015), *P. aeruginosa*: (Cai et al., 2016), *C. crescentus* (Grognot et al., bioRxiv 2020).

Tang and co-workers reported that forward run durations in *C. crescentus* were prolonged (shortened) when the cell migrated up (down) an attractant gradient (Morse et al., 2016). This suggests that the *C. crescentus* modulates its rotational bias during chemotaxis. These results were challenged recently by the work of Grognot and Taute, which is unpublished at the time of our submission.

In the work of Xie et al., 2015, the flagellar motors in *V. alginolyticus* appeared to modulate their reversal frequency without a substantial change in bias. However, recent cryo-ET maps of *V. alginolyticus* motors suggest that the CW conformation is likely stabilized by CheY-P binding (Carroll et al., *eLife*, 2020). The latter result is consistent with the notion that CheY-P binding increases rotational bias, whereas the former is inline with the reviewer’s expectations.

In Cai et al., *mBio*, 2016, the authors assumed that the *P. aeruginosa* cells were tethered by their flagella to the glass surface despite eschewing anti-flagellin antibodies (see Discussion section). As explained in Chawla et al., 2020; Dominick and Wu, 2018; and Lele et al., 2016, these assumptions are not always valid in the absence of antibodies and can lead to errors in distinguishing CW and CCW turns of the motor. This in turn can lead to erroneous measurements of rotational bias.

As the foregoing discussion and response demonstrate, there is plenty of uncertainty in our opinion regarding the modulation of flagellar functions in polar-flagellated species.

b) *H. pylori*'s polar flagella enable locomotion with either direction of rotation, whereas *E. coli*'s peritrichous flagella only enable locomotion in one rotation state. Thus bias modification makes sense for *E. coli*, but not for *H. pylori*.

Our data (Figure 3A-C) argue against reviewer 3’s prediction that *H. pylori* are unlikely to modulate bias. The observation that the reversal frequency exhibits a unimodal dependence on the bias (Figure 4C) likely indicates that the reversal frequency also exhibits a unimodal dependence on CheY-P, similar to *E. coli* (see Figure 6B and Cluzel et al., 2000). Were *H. pylori* to adopt the frequency modulation strategy as suggested by reviewer 3, an increase as well as a decrease in CheY-P would cause a decrease in reversal frequency – meaning both attractant and repellents would induce the same flagellar response and therefore, *H. pylori* would be attracted to repellents as well. This would render the chemotaxis network superfluous.

These findings are impactful since they point to a strategy of chemotaxis signaling in *H. pylori* (and possibly other polar-flagellates) that is presently unknown.

c) The existing H. pylori literature also focusses on the turning frequency.

We agree. We urgently call for a revision of these current approaches in the light of our new findings (Figure 4C). A decrease in reversal frequency can lead to the false impression that the kinase activity has increased when in fact it has decreased and vice-versa (see Discussion section and Figure 6). On the other hand, a change in bias is likely to be more representative of the kinase activity. Given that our study challenges current approaches, we are hopeful that the reviewer will appreciate the impact of our work.

d) To support their assumption, the authors refer to the fact that modifying the turning frequency by concentration does not yield chemotaxis. That is a logical fallacy – the point made in Appendix C of the cited source (Berg's "Random Walks in Biology") is that chemotaxis is achieved by modifying turning frequency in response to changes in chemical concentration, rather than absolute concentrations. It is not a statement on the effectiveness of bias modulation vs turning frequency modulation. If the authors are referring to a different section of the book, they should state explicitly which one.

We agree that this was confusingly presented. Our point was that mere dependence of reversal frequencies on absolute chemical concentrations (or temperatures) is not sufficient for chemotaxis – this is mentioned in the cited source. Yet, current works (Schweinitzer et al., 2008; Rader et al., 2011; Sweeney et al., 2012; Sanders et al., 2013; Collins et al., 2016; Machuca et al., 2017), have quantified the effect of absolute concentrations on the steady-state reversal frequencies to characterize the response to chemicals. Based on our data, we propose that these approaches are inaccurate. We have modified the text to avoid confusion.

e) The "similarity in rotational bias in the two species" is referred to as grounds for assuming similarity in chemotactic strategy. Not only is the bias not similar (*H. pylori* shows similar durations of CCW and CW intervals, while in *E. coli* the typical wildtype CW bias is around ~10-15% (Montrone et al., 1998, Liu et al., 2020 and many others – though Ford et al., 2018 from the authors' lab reports a higher value), the argument also does not hold.

This is a misunderstanding; we never suggested in the original manuscript that the chemotactic strategy is similar between the two species. Our claim was:

“The bias was similar to that observed in *E. coli* (28, 29), suggesting that the basal chemotactic output in the two species is similar.”

The basal chemotactic output refers to kinase activity as well as the rotational bias, and is distinct from chemotaxis (navigation) strategy. We stand by this statement based on our work and that of others, where the bias was ~ 0.3-0.4 in *E. coli*: Block et al., 1982; Block et al., 1983; Segall et al., J Bacteriol, 1986; Sagawa et al., 2014; Stock et al., 1985. We have cited these works.

f) Given the crucial importance of the assumption, solid experimental evidence for it should be presented. That could be e.g. data that show that the bias changes drastically when an attractant/repellent is added.

We believe we have met this requirement.

3) Claim 1 is not novel (see e.g. Constantino et al., Science Advances 2016).

Although we did not claim in the original manuscript that we discovered asymmetric swimming in *H. pylori*, we regret that the description created this impression. We now discuss this finding in the Results.

We draw the reviewer’s attention to the actual data discussed by Constantino and co-workers, a study whose main focus was not on swimming speeds. To distinguish between the two modes in *H. pylori*, the authors attempted to visualize the flagella. However, they were able to demonstrate data for only 1 wild-type cell, where the flagellum is at least partially visible, in the entire manuscript (including supplementary text and video) –. The authors discussed the technical challenges with flagellar visualization, which likely limited sampling. The conclusions were based on their claim that the pole that carried the flagella exhibited rapid changes in contrast during cell movement, which made it unnecessary to view the flagella. However, the authors failed to rigorously support this claim – as mentioned, only 1 cell data was shown where a change in contrast is discernible. In our opinion, no conclusions can be made from such limited data. In the work of Howitt and co-workers, they drew conclusions regarding bias based on a sample size of 8 cells. In both studies, experiments were performed at room temperature.

In contrast, we relied on well-established hydrodynamic models to show that the pusher mode is faster. Our method is high-throughput allowing sample sizes ~ 100 to 900 cells. Finally, we have performed experiments at physiologically relevant temperatures (37°C) for *H. pylori* unlike these other works. Thus, our findings are relevant to the biomedical problems posed by *H. pylori.*

4) Claim 4 is not novel (see Theves et al., 2013 for P. putida).

Theves and co-workers found that their diffusion model for asymmetric swimmers (*P. putida*) was inaccurate as they made an unrealistic assumption that the wait-times were exponentially distributed. However, they did not derive the appropriate expressions for the diffusion coefficients with the more general, Γ distributed wait-times.

In our work, we have shown that *H. pylori* wait time intervals are Γ distributed and not exponentially distributed, hence Theves et al.’s model does not apply. To realistically model the diffuse spread, we developed a model that incorporated the more difficult-to-derive but general form of distributions (Γ distribution). We now clarify this in the main-text. Thus our derivation and models are an improvement over the previous work, as can be seen from the close agreement between simulations and models (Figure 5C).

[Editors’ note: what follows is the authors’ response to the second round of review.]

Revisions:1) The authors state "a very low basal value of the CWbias is disadvantageous as it prevents a response to an attractant stimulus – the cells cannot respond to an attractant if the pre-stimulus bias is ~ 0.", and similarly "But, a very low basal value of the CWbias is disadvantageous as it prevents cells from responding to attractants.". Is this a well-known experimental fact? In which case it may be useful to add a citation. Or do the authors infer this from their own data? In which case I missed something.

The logic here is that if the pre-stimulus (basal) CWbias was already at its minimum value (=0), then further reduction upon stimulation with attractants is not possible. This would inhibit chemotaxis towards attractants. We clarify this in the main text as:

“However, a very low basal value of the CW_bias_ is disadvantageous from a chemotaxis perspective. *H. pylori* appear to respond to attractants by reducing their CW_bias_ (Figure 3C). They would lose their ability to respond to attractants if the pre-stimulus (basal) bias was close to its minimum value (=0)”.

Also:

“Thus, the preference for the faster pusher mode (lower CW_bias_) in *H. pylori* is advantageous as it helps them spread faster (Figure 5D). However, *H. pylori* appear to respond to attractants by reducing their CW_bias_ (Figure 3C). A basal value ~ 0 would mean that the CW_bias_ cannot be reduced further, preventing the cell from responding to attractants.”

2) I think it would be helpful to discuss in the main text the results shown in Figure 5D and E. In particular, for a given asymmetry, the coefficient of diffusion first goes down with CWbias then goes up again, so that for small asymmetries the coefficient of diffusion is similar at low (~0.1) and high (~0.9) CWbias. I guess the diffusion coefficient first goes down because the fraction of time spent in the slow mode increases, but then why does it go up again?

We agree and have provided additional insights regarding the diffusion coefficients in Figure 5D and E. For a symmetric swimmer (*a*=0), the displacement in a run tends to cancel out the displacement during a reversal, which minimizes the diffusivity at CWbias ~ 0.5. The net displacement (and therefore, the diffusivity) increases when the CWbias is less than or greater than 0.5. This is why the diffusivity is high at low CWbias, minimizes at CWbias ~ 0.5, and then increases again at high CWbias. As the asymmetry increases, the net displacement (and therefore the diffusivity) increases when the swimmer prefers the slower mode (CWbias > 0.5). However, the displacement (diffusivity) is much higher when the CWbias is lower – it is more advantageous to spend a greater fraction of the time in the faster mode. We clarify this in the text:

“As shown in Figure 5D, the simulated diffuse spread was low when cells covered similar distances in the forward and backward directions, thereby minimizing net displacement. This tended to occur for swimmers with low *a* values that swam for equal durations in the two directions (CW_bias_ ~ 0.5). For any given *a*, the diffuse spread increased with the net displacement during a run-reversal, for example, when the swimmer preferred the slower mode much more than the faster mode. The net displacement, and hence, the spread tended to be the highest when the cells spent a greater fraction of the time swimming in the faster mode compared to the slower mode.”

3) As I said above, I like the data shown in Figure 4C, which demonstrates that the reversal frequency is likely not a good descriptor for chemotaxis. From what I understand, we're in fact hiding two reversal frequencies in the reversal frequency: the one from puller to pusher, and the one from pusher to puller. As for the CWbias, it tells us about the fraction of time in CW swimming, but not about the time spent in CW swimming. So I'm wondering if there's not a time information missing in the CWbias. Have the authors tried to look at the two reversal frequencies separately? Maybe the frequency of switching from puller to pusher is a more complete description of the data? Or have the authors plotted the CWbias on the x axis, and on the y axis the reversal frequency of slow to fast mode, and the reversal frequency of fast to slow mode? Although I am not asking the authors to redo the analysis, I think if they already have looked at these ways of describing the data, it would be useful for the reader to know it, and potentially to have the corresponding graphs as supplementary material.

We estimated the wait-times for the two modes (τ_pusher_ and τ_pusher_) as a function of the bias as per reviewer request, and have included them in the supplementary information (Appendix 1—figure 3).

4) Figure 3C and 3D: what are the light gray bins? The legend only indicates white for control and dark gray for attractant.

There are only two types of data – white and dark gray. The light gray data showed the overlap between the bars in the histograms. We have fixed this by bringing the dark gray data in front of the white; the overlap can be discerned from horizontal lines within the gray bars in Figure 3C and 3D.